# Ceramides bind VDAC2 to trigger mitochondrial apoptosis

Shashank Dadsena[1,11], Svenja Bockelmann[1,11], John G.M. Mina [1,2,11], Dina G. Hassan[1,3,11], Sergei Korneev[1], Guilherme Razzera[4,5], Helene Jahn[1], Patrick Niekamp[1], Dagmar Müller[1], Markus Schneider[1,6,7], Fikadu G. Tafesse[8], Siewert J. Marrink[9], Manuel N. Melo [4,9] & Joost C.M. Holthuis[1,7,10]

Ceramides draw wide attention as tumor suppressor lipids that act directly on mitochondria to trigger apoptotic cell death. However, molecular details of the underlying mechanism are largely unknown. Using a photoactivatable ceramide probe, we here identify the voltage-dependent anion channels VDAC1 and VDAC2 as mitochondrial ceramide binding proteins. Coarse-grain molecular dynamics simulations reveal that both channels harbor a ceramide binding site on one side of the barrel wall. This site includes a membrane-buried glutamate that mediates direct contact with the ceramide head group. Substitution or chemical modification of this residue abolishes photolabeling of both channels with the ceramide probe. Unlike VDAC1 removal, loss of VDAC2 or replacing its membrane-facing glutamate with glutamine renders human colon cancer cells largely resistant to ceramide-induced apoptosis. Collectively, our data support a role of VDAC2 as direct effector of ceramide-mediated cell death, providing a molecular framework for how ceramides exert their anti-neoplastic activity.

[1] Molecular Cell Biology Division, Department of Biology/Chemistry, University of Osnabrück, 49076 Osnabrück, Germany. [2] School of Science, Engineering and Design, Teesside University, Middlesbrough TS1 3BX, UK. [3] Institute of Environmental Studies and Research, Ain Shams University, Cairo, Egypt. [4] Instituto de Tecnologia Química e Biológica António Xavier, Universidade Nova de Lisboa, Av. da República, 2780-157 Oeiras, Portugal. [5] Departamento de Bioquímica, Centro de Ciências Biológicas, Universidade Federal de Santa Catarina, Florianópolis, Brazil. [6] Plant Physiology Division, Department of Biology/Chemistry, University of Osnabrück, 49076 Osnabrück, Germany. [7] Center for Cellular Nanoanalytics, Osnabrück University, Artilleriestraße 77, 49076 Osnabrück, Germany. [8] Molecular Microbiology and Immunology, Oregon Health & Science University, Portland, OR 97239, USA. [9] Groningen Biomolecular Sciences and Biotechnology Institute and Zernike Institute for Advanced Materials, University of Groningen, Nijenborgh 7, 9747 AG Groningen, The Netherlands. [10] Membrane Biochemistry and Biophysics, Bijvoet Center and Institute of Biomembranes, Utrecht University, 3584 CH Utrecht, The Netherlands. [11] These authors contributed equally: Shashank Dadsena, Svenja Bockelmann, John G. M. Mina, Dina G. Hassan. Correspondence and requests for materials should be addressed to J.G.M.M. (email: j.mina@tees.ac.uk) or to M.N.M. (email: m.n.melo@itqb.unl.pt) or to J.C.M.H. (email: holthuis@uos.de)

Sphingolipids are essential components of eukaryotic membranes that participate in a broad range of cellular processes by controlling vital physical membrane properties and as signaling molecules in response to physiological cues and stresses[1–3]. Notably ceramides, the central intermediates of sphingolipid metabolism, have emerged as key mediators of anti-proliferative and tumor suppressive cellular programs such as apoptosis, mitophagy, cell cycle arrest, and senescence[4,5]. Multiple stress stimuli, including tumor necrosis factor α (TNFα)[6,7], ionizing radiation[8,9], and chemotherapeutic drugs[10,11], cause a rise in ceramide levels through stimulation of de novo ceramide synthesis, activation of sphingomyelin hydrolysis, or both. Interventions that suppress ceramide accumulation render cancer cells resistant to these stress-inducing agents, indicating that ceramides are bona fide anti-proliferative and pro-apoptotic signaling molecules. While these findings raised considerable interest in targeting sphingolipid-metabolizing enzymes for cancer therapy[5,12,13], the mechanisms by which ceramides execute their tumor-suppressive activities are incompletely understood.

Mitochondria serve a central role in apoptosis induced by stress stimuli. The mitochondria from cancer cells are often resistant to induction of mitochondrial outer membrane permeabilization (MOMP), a point of no return in the intrinsic pathway of apoptosis. MOMP allows the passage of intermembrane space proteins such as cytochrome $c$ to activate caspases, a family of cysteine proteases responsible for executing an ordered destruction of the cell[14]. MOMP is controlled by pro- and anti-apoptotic members of the B-cell lymphoma 2 (Bcl-2) protein family, which collectively determine the balance between cell death and survival[15,16]. The main function of the anti-apoptotic Bcl2 proteins is to counter the pro-apoptotic activities of the Bcl-2 proteins Bax and Bak, which directly mediate MOMP by creating proteolipid pores responsible for cytochrome $c$ release[17,18]. Several reports have indicated that ceramides can trigger MOMP by modulating the activity of kinases or phosphatases implicated in controlling Bcl-2 protein function. In cells, elevated ceramide levels have been shown to inhibit phosphoinositide-3-kinase (PI3K) and Akt/PBK signaling, resulting in dephosphorylation and subsequent activation of pro-apoptotic Bcl-2-family protein Bad[19,20]. Short-chain ceramides can bind and stimulate protein phosphatase 2A (PP2A), which dephosphorylates and inactivates the anti-apoptotic protein BCL2[21,22].

Other studies revealed that ceramides can also act directly on mitochondria to trigger MOMP and apoptotic cell death[23]. For instance, mitochondrial targeting of a bacterial sphingomyelinase to generate ceramides in mitochondria or directing CERT-mediated ceramide transport to mitochondria induces cytochrome $c$ release and apoptosis[24,25]. In addition, ER-like membranes associated with isolated mitochondria appear to produce sufficient amounts of ceramides to enable a transient passage of cytochrome $c$ across the outer membrane[26]. However, the underlying mechanisms remain to be established. Interestingly, ceramides have been shown to form pores in model bilayers as well as in the outer membrane of isolated mitochondria that are large enough to mediate passage of cytochrome $c$[27,28]. Formation of ceramide channels does not rely on any particular protein but is disrupted by anti-apoptotic Bcl-2 proteins[29]. It has also been suggested that ceramides accumulating in the mitochondrial membrane of mammalian cells upon irradiation form or stabilize microdomains that serve as platforms into which Bax inserts and assembles into an active pore[30]. While ceramides have been proposed to cooperate directly with Bax in the assembly of cytochrome $c$ conducting channels[30,31], other experiments with isolated mitochondria suggest that metabolic conversion of ceramides into sphingosine-1-phosphate and hexadecenal is necessary to facilitate Bax/Bak activation leading to MOMP[32].

In this study, we present evidence for an alternative mechanistic view, namely that ceramides mediate their pro-apoptotic activity at least in part by interacting directly and specifically with the voltage-dependent anion channel VDAC2, a mitochondrial platform for Bax/Bak translocation[33–35]. Identification of VDAC2 as an effector of ceramide-mediated cell death provides new opportunities for exploiting the therapeutic potential of ceramides as tumor suppressor lipids.

## Results

**A chemical screen for ceramide-binding proteins yields VDACs.** To identify proteins involved in ceramide-mediated stress signaling and apoptosis, we used a bifunctional ceramide analog carrying a photoactive diazirine and clickable alkyne group in its $N$-linked acyl chain (pacCer, Fig. 1a)[36]. Total membranes from human HeLa cells were incubated with pacCer-containing liposomes, subjected to UV crosslinking and click reacted with Alexa Fluor 647 azide (AF647-N$_3$). In-gel fluorescence (IGF) analysis revealed a subset of membrane-bound proteins with affinity for the pacCer probe, which included a mitochondria-associated protein of ~33 kDa that was prominently photolabeled (Fig. 1b, Supplementary Fig. 1). As ceramide exerts its apoptogenic activity in mitochondria[8,24,25], we set out to identify the 33 kDa candidate ceramide-binding protein (CBP). To this end, mitochondria were photolabeled with pacCer and then click reacted with a PEG-based reagent containing an azide, a biotin and a TAMRA fluorophore as functional groups (Fig. 1c). Next, pacCer-crosslinked proteins were isolated using NeutrAvidin agarose and visualized by IGF (Fig. 1d). The fluorescent 33 kDa protein band was cut from the gel, trypsin-digested, and identified by LC-MS/MS analysis as the voltage-dependent anion channel isoforms VDAC1 and VDAC2 (Supplementary Table 1). In line with the MS data, pretreatment of HeLa cells with VDAC1 and VDAC2-targeting siRNAs effectively depleted the fluorescent 33 kDa protein band from pacCer-labeled and Alexa click-reacted mitochondria (Fig. 1e, f). The photolabeled 33 kDa protein band also cross-reacted with both anti-VDAC1 and anti-VDAC2 antibodies (Fig. 1g, h). In contrast, VDAC3 and TOM40—an outer mitochondrial membrane (OMM) channel protein with a cellular copy number close to that of VDACs[37]—both lacked affinity for pacCer, indicating that VDAC1 and VDAC2 are genuine mitochondrial CBPs.

**MD simulations uncover a ceramide-binding site on VDACs.** To search for a putative ceramide-binding site on VDAC1 and VDAC2, we performed coarse-grain molecular dynamics (CG-MD) simulations using the Martini model[38–40]. Main simulations were performed with VDAC channels at an aggregate time of 1.23 ms (Supplementary Table 2)—only attainable using CG-MD. A well-resolved structure of mouse VDAC1 (PDB: 4C69)[41] was used as a base template. An available structure of VDAC2 from zebrafish (PDB:4BUM)[42] showed almost perfect structural identity to VDAC1 (1.7 Å barrel backbone RMSD). From the assumption of identical secondary structure we mutated VDAC1 side chains to the mouse sequences of VDAC2 and VDAC3 to obtain all three isoforms for comparison. A bilayer mimicking the OMM[43] was constructed with ~630 lipids. To the outer leaflet of this bilayer, 16 molecules of C$_{16}$-ceramide were added. In agreement with the photolabeling data, simulations revealed that VDAC1 and VDAC2, but not VDAC3, have a binding site for ceramide buried in the membrane interior on one side of the barrel wall, comprising β-strands 3–5 (Fig. 2a). This site harbors a uniquely positioned glutamate (Glu) residue in the transmembrane region of β-strand 4—Glu73 in VDAC1 and Glu84 in VDAC2—that faces the bilayer's hydrophobic core

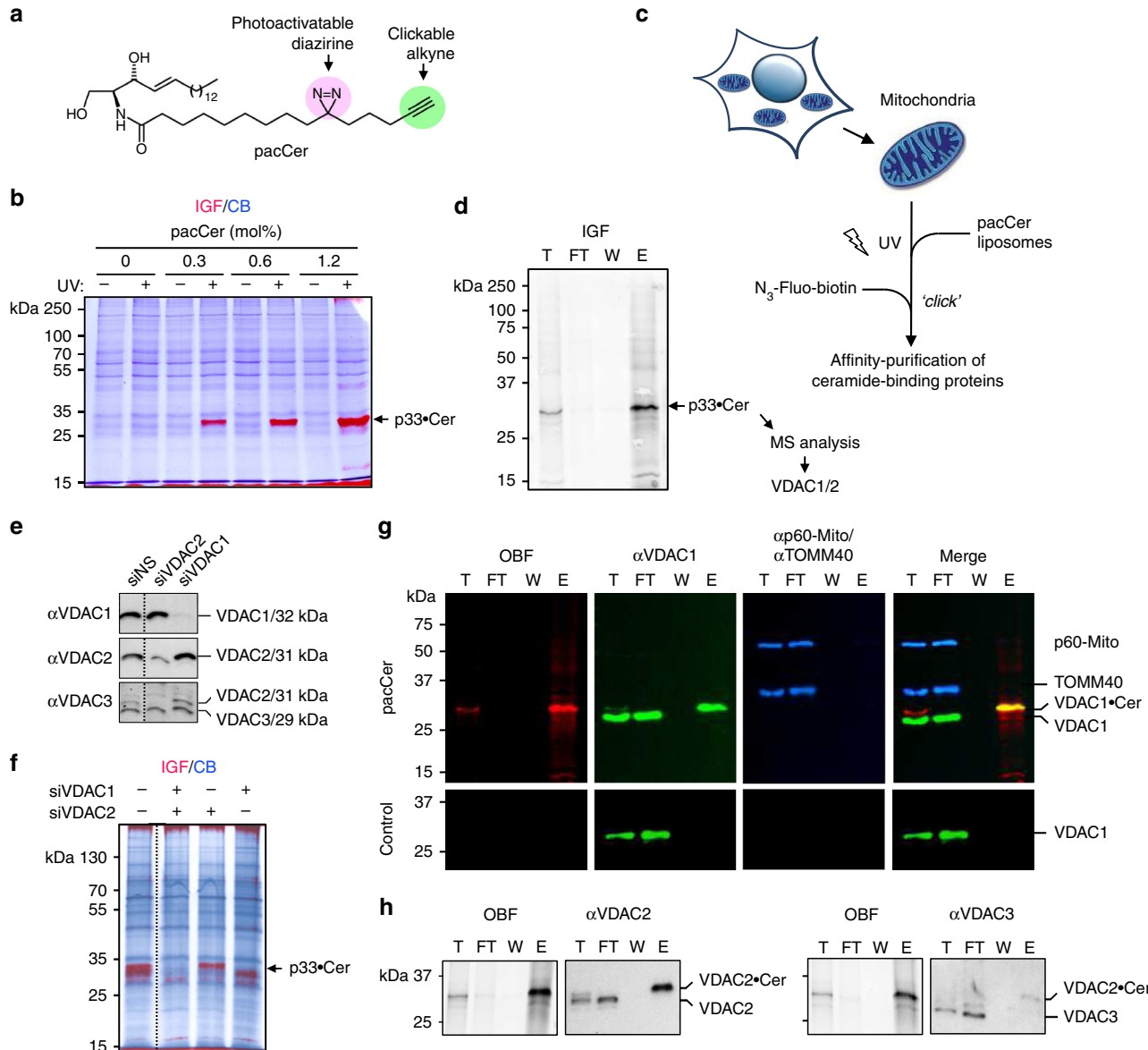

**Fig. 1** A chemical screen for mitochondrial ceramide-binding proteins yields VDAC1 and -2. **a** Structure of the photoactive and clickable C$_{15}$-ceramide analog, pacCer. **b** Mitochondria isolated from HeLa cells were incubated with liposomes containing increasing amounts of pacCer, UV irradiated, and then click reacted with AF647-N$_3$. Samples were processed for SDS-PAGE, subjected to in-gel fluorescence (IGF, red), and stained with Coomassie blue (CB, blue). p33•Cer denotes a prominently photolabeled protein band of ~33 kDa. **c** Strategy for the identification of p33•Cer. **d** p33•Cer was purified from pacCer-labeled and TAMRA/biotin click-reacted mitochondria using NeutrAvidin-beads, imaged by IGF, excised from the gel, digested by trypsin, and then identified by LC-MS/MS. Data from two independent experiments revealed that p33•Cer corresponds to VDAC1 and VDAC2. **e** Specificity of anti-VDAC antibodies was validated by immunoblotting of mitochondria isolated from HeLa cells treated with non-silencing (siNS) or VDAC-targeting siRNAs (siVDAC1, siVDAC2). Note that the anti-VDAC3 antibody cross-reacts with VDAC2. **f** Mitochondria isolated from siVDAC1/2-treated HeLa cells were photolabeled with pacCer, click-reacted with AF647-N$_3$, and subjected to IGF analysis followed by CB staining. **g** Fractions obtained during affinity purification of p33•Cer were subjected to SDS-PAGE, transferred on nitrocellulose, analyzed by on-blot-fluorescence (OBF) and probed with antibodies against VDAC1, TOM40, and p60-Mito. **h** Fractions obtained during affinity purification of p33•Cer were processed as in **g** and probed with anti-VDAC2 and anti-VDAC3 antibodies. T total mitochondria extract, FT flow-through, W wash, E eluate

(Fig. 2b). In its deprotonated state, this residue seemed to promote direct contact with the ceramide head group (Fig. 2c; Supplementary Information Videos 1 and 2). In VDAC3, which does not bind ceramide (Figs. 1h and 2a), the bilayer-facing Glu residue is replaced by a glutamine (Gln73; Fig. 2b). Substitution of Gln for Glu73 in VDAC1 or Glu84 in VDAC2 strongly reduced the ceramide occupancy and residence time at the binding sites (Fig. 2d, e). Protonation of the bilayer-facing Glu also greatly diminished ceramide-binding (Fig. 3a, b) while substitution of a

deprotonated aspartate for the Glu residue retained ceramide binding (Supplementary Fig. 2). This indicates that a negative charge on the membrane-buried Glu residue is critical for ceramide binding.

In line with a previous computational study[44], we also found a number of binding sites for cholesterol. These displayed no overlap with the ceramide-binding site (Fig. 2d). To exclude the possibility that competition with ceramide prevented cholesterol to occupy the ceramide-binding site, additional simulations were

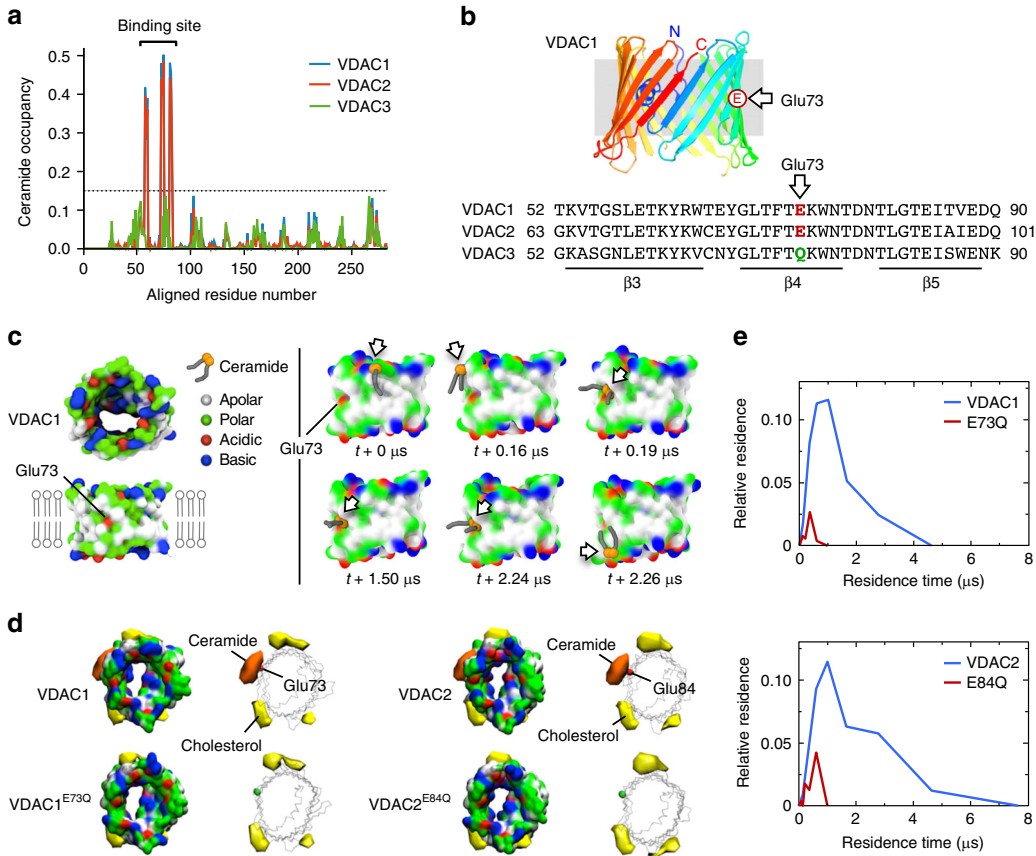

**Fig. 2** MD simulations uncover a putative ceramide-binding site on VDAC1 and -2. **a** Ceramide head group contact occupancy of mouse VDAC1, VDAC2, and VDAC3 in an OMM model containing 5 mol% ceramide, with 1.0 corresponding to a ceramide contact during the entire simulation time. A threshold of 15% occupancy, based on the occupancies of non-binding site residues, is indicated by a dotted blue line. VDAC1 and VDAC2 have a clear ceramide-binding site, comprising residues 58–62, 71–75, and 81–85; this site is lacking in VDAC3. **b** Sequence alignment revealing the position of a bilayer-facing Glu residue in VDAC1 (E73) and VDAC2 (E84), which is replaced by Gln in VDAC3 (Q73). **c** Stills from an MD simulation, showing the approach and binding of a ceramide molecule to VDAC1 in close proximity of the bilayer-facing Glu residue in its deprotonated state. Protein surface colors mark polar (green), apolar (white), cationic (blue), or anionic (red) residues. **d** Space-filling and wireframe models of VDAC1, VDAC1[E73Q], VDAC2, and VDAC2[E84Q] with deprotonated E73/E84. Indicated are the volumes for which there is ceramide occupancy greater than 10% (orange) or cholesterol occupancy greater than 20% (yellow). **e** Distribution of the durations of ceramide contacts with VDAC1, VDAC1[E73Q], VDAC2, and VDAC2[E84Q] at the preferred binding site as in **d**. The y-axis indicates the fraction of the total system time spent in binding events of the duration indicated by x. Summing all points' y-values yields the fraction of total simulation time when ceramide was bound

performed in the absence ceramide. Also in those simulations no cholesterol binding near the membrane-facing Glu was observed (Supplementary Fig. 3). We observed zero-specific binding events between phosphatidylcholine (PC) and VDACs in the OMM mimics. Yet when VDAC1 was simulated in a bilayer of 100% dimyristoyl-phosphatidylcholine (DMPC) following the setup of a previous atomic resolution simulation analysis[45], contacts of the PC head group with the bilayer-facing deprotonated Glu residue could occasionally be observed (Supplementary Fig. 4a). However, these encounters were much shorter-lived (≤5 ns; Supplementary Fig. 4b) and extremely rare in comparison to those involving ceramide, which displayed average residence times of 0.8 and 1.2 μs for VDAC1 and VDAC2, respectively (Supplementary Fig. 4c). In sum, CG-MD simulations revealed that VDAC1 and VDAC2 each harbor a binding site for ceramide, with a charged Glu residue buried in the membrane interior serving a critical role in ceramide binding.

**Ceramide binding relies on a membrane-facing glutamate.** We next sought to verify the relevance of the membrane-facing Glu residue in VDACs for ceramide binding. To this end, human

VDAC1 and VDAC2 and the mutant channels VDAC1[E73Q] and VDAC2[E84Q] were produced recombinantly in *E. coli* and then reconstituted in egg PC liposomes (Supplementary Fig. 5). Density gradient fractionation analysis revealed that reconstitution efficiencies of wild type and mutant channels were practically indistinguishable. The reconstituted channels were then subjected to photolabeling with pacCer and bifunctional analogs of diacylglycerol (pacDAG), PC (pacPC), phosphatidylethanolamine (pacPE), and cholesterol (pacChol; Supplementary Fig. 6). VDAC1 and VDAC2 could be efficiently and reproducibly photolabeled with pacCer, pacPC, and pacChol, but not with pacDAG or pacPE (Fig. 4). In agreement with the simulations, replacing the membrane-facing Glu with Gln virtually abolished labeling of both channels with pacCer and pacPC, whereas labeling with pacChol was not or only slightly affected (Figs. 4 and 5a). Moreover, reducing the pH from 7 to 5 caused a significant reduction in E73-dependent photolabeling of VDAC1 with pacCer (Fig. 3c, d). This suggests that ceramide binding is critically dependent on the protonation state of the membrane-exposed Glu, as predicted by the simulations.

The pronounced labeling of the wild-type channels with pacPC was somewhat unexpected as simulations revealed that specific

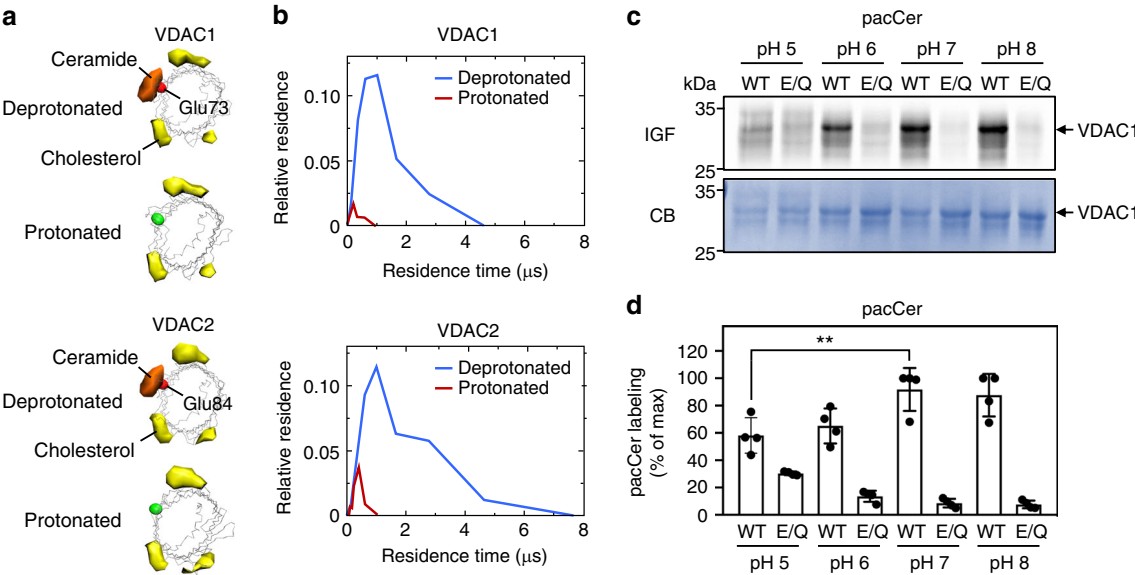

**Fig. 3** Ceramide binding by VDACs relies on the protonation state of the bilayer-facing Glu. **a** Space-filling and wireframe models of VDAC1 and VDAC2 simulated with the bilayer-facing Glu residue in a protonated or deprotonated state. Indicated are the volumes for which there is ceramide occupancy greater than 10% (orange) or cholesterol occupancy greater than 20% (yellow). **b** Distribution of the durations of ceramide contacts with VDAC1 and VDAC2 at the preferred binding site as in **a**. The y-axis indicates the fraction of the total system time spent in binding events of the duration indicated by x. Summing all points' y-values yields the fraction of total simulation time when ceramide was bound. **c** Human VDAC1 and VDAC1[E73Q] were produced in *E. coli*, purified, reconstituted in liposomes, and then photolabeled at the indicated pH with pacCer added from an ethanolic stock. Samples were click-reacted with AF647-N$_3$, subjected to SDS-PAGE, and analyzed by IGF and CB staining. **d** Quantitative analysis of relative pacCer photolabeling efficiencies of reconstituted VDAC1 treated as in **c**. Data are means ± s.d.; n = 4; \*\*p < 0.01 by two-tailed paired t-test. Source data

encounters of PC molecules with the membrane-facing Glu are relatively rare. This discrepancy could be due to the irreversible nature of photoaffinity labeling in combination with the major difference in timescales between CG-MD simulations (μs range) and photoaffinity labeling (second range). While our data revealed no obvious overlap between the ceramide and cholesterol binding sites on VDACs, a recent photolabeling study mapped the bilayer-facing Glu to a major cholesterol binding pocket on VDAC1[46]. In contrast to the present work, the latter study was performed on VDAC1-containing bicelles with cholesterol probes that carry the photoactive diazirine in the aliphatic tail or at C7. Thus, additional work will be necessary to resolve the discrepancy between the simulations and photolabeling studies of cholesterol binding to VDACs.

As aliphatic diazirines display photochemical preference for nucleophilic amino acids[47], the deprotonated side chain of the membrane-buried Glu in VDACs may provide a site of insertion for the diazirine in pacCer, which may diffuse a short distance to reach such a nucleophile. Therefore, we next labeled VDACs with pacCer in the presence of excess C$_{16}$-ceramide. Labeling by pacCer was progressively reduced by C$_{16}$-ceramide when added in 3- to 27-fold excess (Fig. 5b), arguing against the idea that pacCer labeling of VDACs is primarily driven by affinity of the aliphatic diazirine for the negatively charged Glu. Excess C$_{16}$-ceramide did not affect labeling of VDACs with pacChol. In line with the simulations, these data indicate that ceramides bind VDACs at the membrane-facing Glu residue. To verify that this concept not only holds for recombinant channel proteins in synthetic bilayers but also for their native counterparts in mitochondrial membranes, we made use of the carboxyl-modifying reagent dicyclohexylcarbodiimide (DCCD). This hydrophobic compound irreversibly reacts with a number of integral membrane proteins through covalent modification of membrane-embedded Asp or Glu residues and was previously shown to modify Glu73 in VDAC1[48]. Pretreatment of

mitochondria with DCCD selectively abolished photolabeling of the 33 kDa protein band with pacCer (Supplementary Fig. 7), hence providing complementary proof that the membrane-facing Glu residue in VDAC1 and VDAC2 is part of an authentic ceramide-binding site.

**Loss of VDAC2 disrupts ceramide-induced apoptosis.** VDAC channels are active participants in the cytosolic release of apoptogenic proteins from mitochondria. VDAC2 serves as a platform for the mitochondrial recruitment of pro-apoptotic Bcl-2 proteins Bak and Bax[33–35], which can commit cells to death by permeabilizing the OMM for cytochrome *c*[17,18]. In response to various apoptotic stimuli, VDAC1 forms oligomers and participates in the assembly of a cytochrome *c*-conducting pore[49,50]. Identification of a ceramide-binding site on VDAC1 and VDAC2 raised the question whether ceramides exert their apoptotic activity by interacting with these proteins. To address this, we employed an engineered ceramide transfer protein equipped with an OMM anchor, mitoCERT (Fig. 6a). We previously demonstrated that human HCT116 colon cancer cells expressing mitoCERT undergo Bax-dependent apoptosis by mistargeting newly synthesized ER ceramides to mitochondria (Fig. 6b)[25]. This led us to determine the impact of VDAC removal on mitoCERT-induced apoptosis in HCT116 cells. Loss of VDAC1, VDAC2, or both was verified by immunoblotting and IGF analysis of mitochondria photolabeled with pacCer (Supplementary Fig. 8). Expression of mitoCERT in wild-type HCT116 cells triggered apoptosis, as indicated by cleavage of caspase substrate PARP1 (Fig. 6c). No PARP1 cleavage was observed in cells expressing a mitoCERT variant that lacked the ceramide transfer or START domain, mitoCERTΔSTART. Loss of VDAC1 had no effect on the ability of mitoCERT to induce PARP1 cleavage. In contrast, VDAC2 removal rendered cells partially resistant to mitoCERT-induced PARP1 cleavage, especially in the absence of VDAC1 (Fig. 6c).

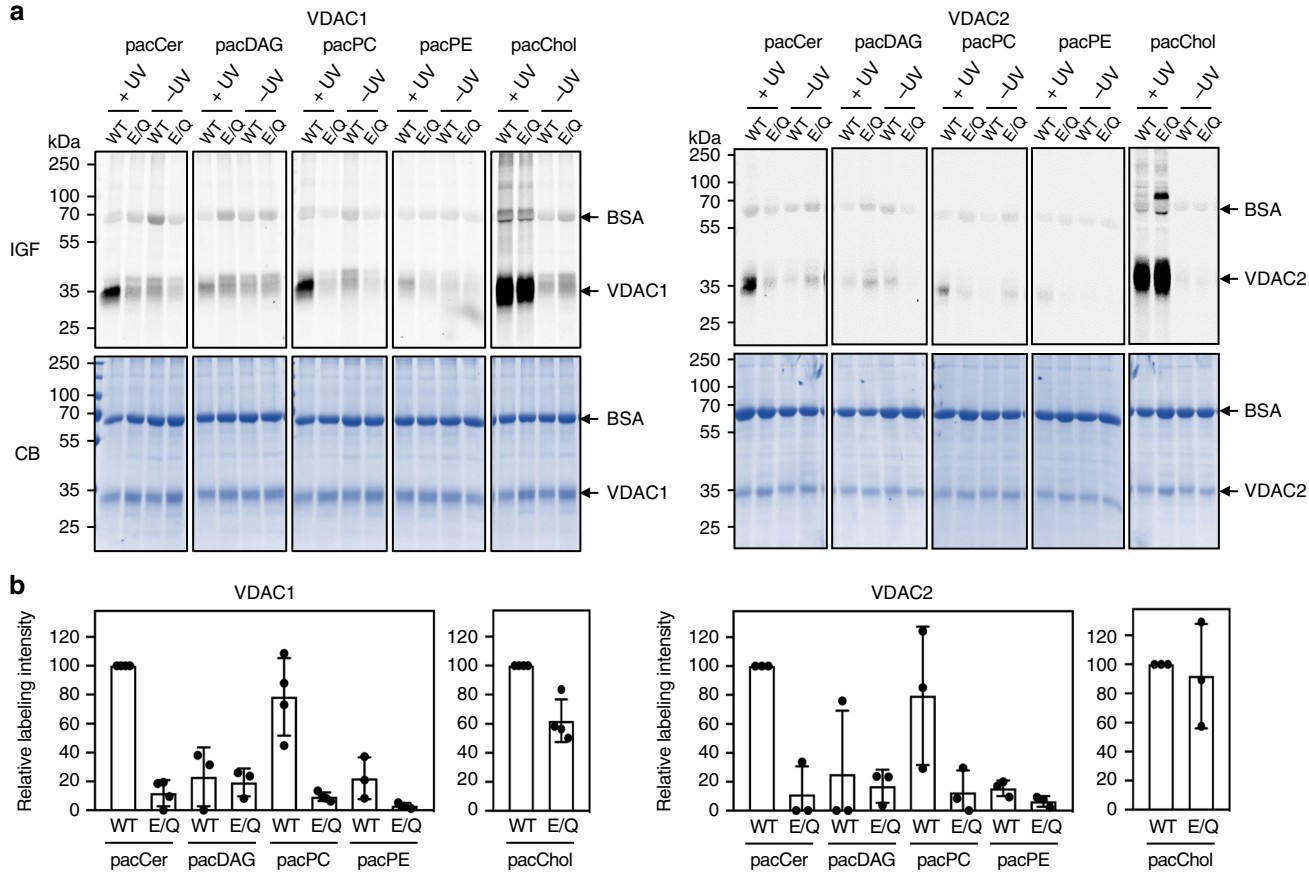

**Fig. 4** The bilayer-facing Glu is a critical determinant of pacCer photolabeling of VDACs. **a** Human VDAC1, VDAC1[E73Q], VDAC2, and VDAC2[E84Q] were produced in *E. coli*, purified, reconstituted in liposomes, and incubated for 30 min at 37 °C with liposomes containing 1 mol% of pacCer or photoactive and clickable analogs of diacylglycerol (pacDAG), phosphatidylcholine (pacPC), phosphatidylethanolamine (pacPE), or cholesterol (pacChol). Samples were UV irradiated and then click-reacted with AF647-N₃, subjected to SDS-PAGE, and analyzed by IGF and CB staining. **b** Quantitative analysis of relative labeling efficiencies of reconstituted VDAC1, VDAC1[E73Q], VDAC2, and VDAC2[E84Q] with pacLipids as indicated in **a**. Data are means ± s.d.; $n \geq 3$. Source data

**Glu84 in VDAC2 is critical for ceramide-induced apoptosis.** We next analyzed individual wild type and ceramide-binding defective VDAC channels for their ability to support mitoCERT-induced apoptosis. To this end, HCT116 VDAC1/2 double KO cells were stably transduced with haemagglutinin (HA)-tagged VDAC1, VDAC1[E73Q], VDAC2, or VDAC2[E84Q] (Fig. 7a). Mitochondrial localization of the tagged channels was confirmed by immunofluorescence microscopy (Supplementary Fig. 9). Heterologous expression of VDAC2, but not VDAC1, restored mitoCERT-induced apoptosis in the double KO cells, as evidenced by PARP1 cleavage and proteolytic activation of caspase-3, the principal executioner caspase in apoptotic cells (Fig. 7a–c). This confirmed that VDAC2, unlike VDAC1, is a key player in ceramide-mediated cell death. Strikingly, replacing Glu84 with Gln in VDAC2 greatly reduced its ability to support mitoCERT-induced apoptosis in the double KO cells (Fig. 7a–c), indicating that cell death triggered by a rise in mitochondrial ceramides requires a ceramide-binding competent form of VDAC2. While VDAC1 or VDAC2 removal had no major impact on cellular Bax levels, cells lacking VDAC2 had strongly reduced Bak levels (Supplementary Fig. 10a), consistent with the unique role of VDAC2 in stabilizing Bak[33]. Reintroducing VDAC2 or VDAC2[E84Q] in VDAC1/2 double KO cells in each case fully restored Bak levels to those in wild-type cells (Supplementary Fig. 10b), indicating that an intact ceramide-binding site is dispensable for VDAC2-mediated stabilization of Bak. This notion is supported by the finding that the Bak-stabilizing activity of VDAC2 requires isoform-specific sequence motifs located outside of the region involved in ceramide binding[33].

## Discussion

The current study identified a role of VDAC2 as a direct and specific effector of ceramide-mediated cell death. This function critically relies on a uniquely positioned charged Glu residue that mediates direct contact with the ceramide head group in the bilayer interior, potentially driven by electrostatic attraction. We speculate that an amide-containing backbone combined with a small polar head group renders ceramide the preferred lipid binding partner of VDAC isoforms containing the membrane-buried Glu residue. Although it is energetically unfavorable for charged residues to face the bilayer's hydrophobic core, recent work revealed that the p$K$a of Glu73 in VDAC1 is closely tuned to the physiological pH of the cytosol (p$K$a ~7.4)[51]. This implies that under stress-free conditions, the membrane-buried Glu is in its deprotonated fully charged state at least a significant amount of time. Interestingly, ceramide binding to the late endosomal protein LAPTM4B depends on a membrane-embedded aspartate[52]. We anticipate that also other proteins with charged acidic residues in their membrane spans may have affinity for ceramide and potentially participate in ceramide-operated signaling pathways.

While ceramide-induced apoptosis requires Bax[8,24,25], a recent study revealed that VDAC2 specifies Bax recruitment to

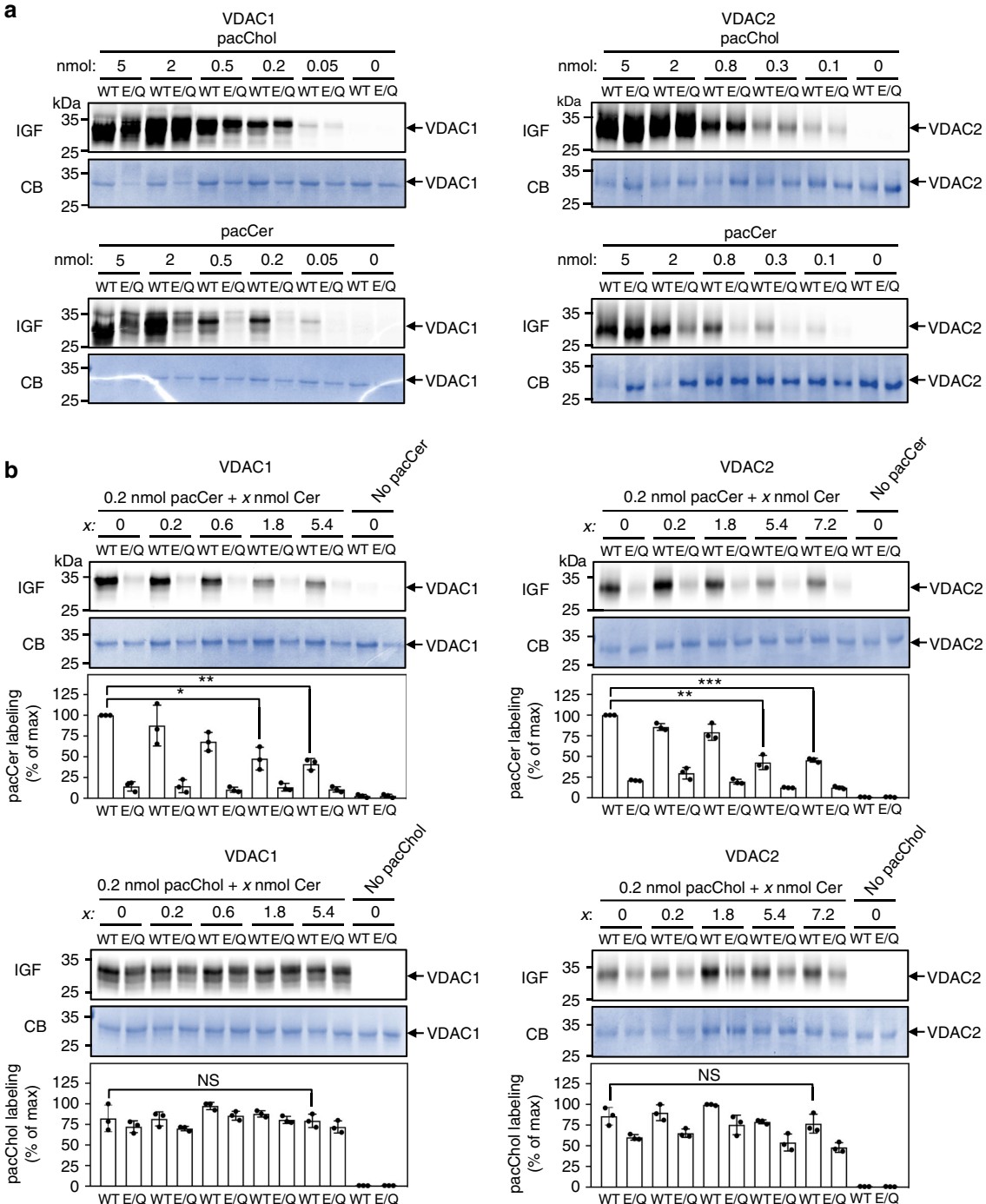

**Fig. 5** Competitive inhibition of pacCer photolabeling of VDACs by $C_{16}$-ceramide. **a** VDAC1, VDAC1$^{E73Q}$, VDAC2, and VDAC1$^{E84Q}$ proteoliposomes were photolabeled with the indicated amount of pacCer or pacChol added from ethanolic stocks. Next, samples were click-reacted with AF647-$N_3$, subjected to SDS-PAGE, and analyzed by IGF and CB staining. **b** VDAC1, VDAC1$^{E73Q}$, VDAC2, and VDAC1$^{E84Q}$ proteoliposomes were photolabeled with 0.2 nmol pacCer or pacChol in the presence of the indicated amounts of natural $C_{16}$-ceramide (Cer) added from ethanolic stocks and then processed as in **a**. Relative labeling efficiencies were quantified and expressed as % of control (0.2 nmol pacCer or pacChol in the absence of Cer). Data are means ± s.d.; $n = 3$; *$p < 0.05$, **$p < 0.01$, and ***$p < 0.001$ by two-tailed paired $t$-test. Source data

mitochondria and concomitantly ensures Bax inhibition by mediating its retrotranslocation into the cytosol[34]. By establishing a dynamic equilibrium between mitochondrial and cytosolic Bax pools, this VDAC2-dependent shuttling is ideally suited for regulation by pro- and anti-apoptotic cues. Our present findings suggest that ceramide binding to VDAC2 may commit cells to death by blocking Bax retrotranslocation. This concept is distinct from previous models postulating that ceramides accumulating in

the OMM: (i) self-assemble into cytochrome $c$-conducting channels[27,28]; (ii) form lipid macrodomains into which Bax inserts and functionalizes as a pore[30,31]; (iii) affect mitochondrial shape to facilitate Bax recruitment and apoptosis[53]; (iv) require metabolic conversion to gain apoptogenic activity[32].

How ceramide binding tips the balance in VDAC2-mediated shuttling of Bax to trigger mitochondrial apoptosis remains to be established. Our simulations and photoaffinity experiments

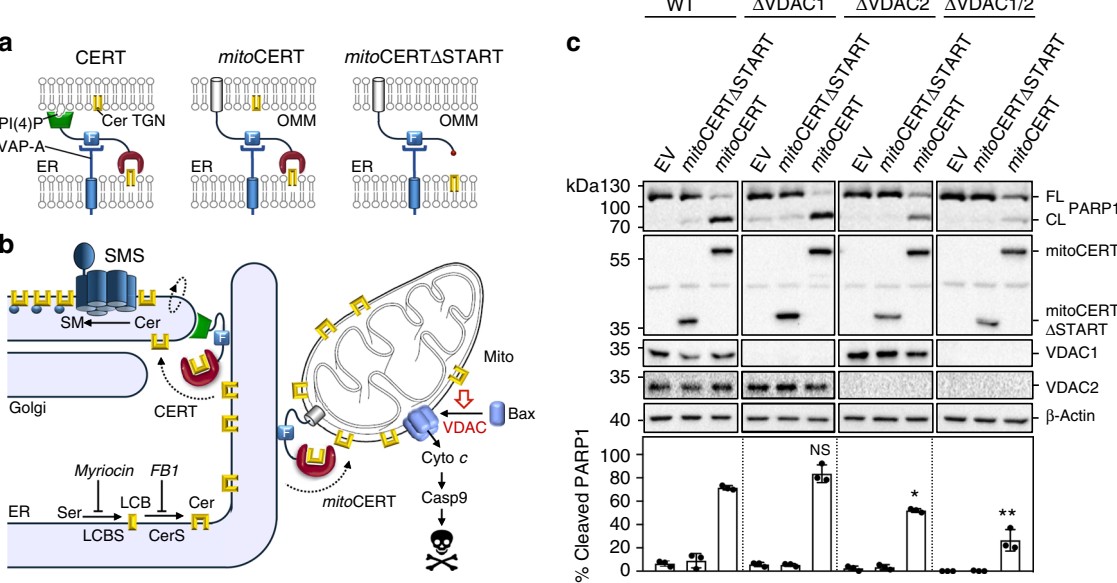

**Fig. 6** VDAC2 removal disrupts ceramide-induced apoptosis. **a** Schematic outline of ceramide transfer protein CERT, mitoCERT, and mitoCERTΔSTART. MitoCERT was created by swapping the Golgi-targeting pleckstrin homology domain of CERT against the OMM anchor of AKAP1. Removal of the ceramide transfer or START domain yielded mitoCERTΔSTART. All three proteins bind the ER-resident protein VAP-A via their FFAT motif (F). Cer ceramide, PI(4)P phosphatidylinositol-4-phosphate, TGN *trans*-Golgi network. **b** Ceramides (Cer) are synthesized through N-acylation of long chain bases (LCB) by ceramide synthases (CerS) on the cytosolic surface of the ER and require CERT-mediated transfer to the Golgi for metabolic conversion into sphingomyelin (SM) by a Golgi-resident SM synthase (SMS). Expression of mitoCERT causes a diversion of this biosynthetic ceramide flow to mitochondria, triggering Bax-dependent apoptosis[11]. **c** Wild type (WT), VDAC1-KO (ΔVDAC1), VDAC2-KO (ΔVDAC1), and VDAC1/2 double KO (ΔVDAC1/2) human colon cancer HCT116 cells were transfected with empty vector (EV), Flag-tagged mitoCERT, or Flag-tagged mitoCERTΔSTART. At 24 h post transfection, cells were processed for immunoblotting with antibodies against PARP1, the Flag-epitope, VDAC1, VDAC2, and β-actin. The percentage of PARP1 cleavage was quantified. Data are means ± s.d.; $n = 3$; *$p < 0.05$ and **$p < 0.01$ by two-tailed paired $t$-test. Source data

indicate that ceramide binding to VDACs is pH sensitive and controlled by the protonation state of the membrane-buried Glu. Interestingly, acidification has been shown to promote association of two VDAC1 monomers into a dimer[51]. Assembly of a high-affinity dimer relies on protonation of the membrane-facing Glu (E73) and likely involves hydrogen bonding between this residue and a serine (Ser43) at the dimer interface[51]. VDAC oligomerization has been implicated as a component of the mitochondrial pathway of apoptosis[49,50] and may be part of the mechanism by which VDAC2 stabilizes the mitochondrial pool of Bax in response to apoptotic stimulation[34]. In view of our present findings, a prospect that merits further investigation is whether binding of ceramide to the charged Glu residue lowers the threshold for VDAC oligomerization at neutral pH. Ceramide binding to VDACs may also influence interactions with other proteins, as the membrane-facing Glu residue is critical for association of VDAC1 with hexokinase I[54,55]. VDAC-bound hexokinases are thought to play a pivotal role in promoting cell growth and survival in rapidly growing, hyperglycolytic tumors[56,57]. Consequently, our current study establishes a molecular framework to unravel how ceramides execute their tumor suppressor functions.

## Methods

**Reagents**. 1,2-Dioleoyl-*sn*-glycero-3-phosphocholine (DOPC), 1,2-dioleoyl-*sn*-glycero-3-phosphoethanol-amine (DOPE), and L-α-phosphatidylcholine from chicken egg (egg PC), 1-palmitoyl-2-{12-[(7-nitro-2-1,3-benzoxadiazol-4-yl) amino]dodecanoyl}-*sn*-glycero-3-phosphocholine (NBD-PC) and $C_{16}$-ceramide (d18:1/16:0) were purchased from Avanti Polar Lipids. Alexa Fluor 647-$N_3$ (AF647-$N_3$) and Biotin-$N_3$ were from Thermo Fischer Scientific, and TAMRA-Biotin-$N_3$ from Click Chemistry Tools. The photoactive and clickable *trans*-sterol probe (pacChol) was from Sigma-Aldrich. A 15 carbon-long fatty acid containing a photoactivatable diazerine and clickable alkyne group, pacFA, was synthesized in three steps from commercially available educts[36]. Next, pacFA was coupled to D-

*erythro*-sphingosine (Enzo Biochem) using a combination of 1-ethyl-3-(3-dime-thylaminopropyl)carbodiimide (EDCI) and hydroxybenzotriazole (HOBT) as condensing reagents, yielding the photoactivatable and clickable C15-ceramide analog, pacCer (85% overall yield). pacPC was synthesized starting from 1-oleoyl-2-hydroxy-*sn*-glycero-3-phosphocholine (Avanti Polar Lipids) and pacFA under the action of N,N-dicyclohexylcarbodiimide (DCC) and 4-dimethylaminopyridine (DMAP) with satisfactory yield (39%). pacDAG was synthesized in three steps starting from 1-oleoyl-*sn*-glycerol (Santa Cruz Biotechnology). First, the primary HO-group was protected with the triphenylmethyl protecting group (trityl-chloride/pyridine; 92% overall yield). The 1-acyl-3-trityloxy-glycerol obtained was coupled with the pacFA using EDCI/DMAP activation (58% overall yield). The final deprotection step was achieved using trifluoroacetic acid (TFAA) to generate pacDAG (28% overall yield). pacPE was synthesized in three steps starting from 1-oleoyl-2-hydroxy-*sn*-glycero-3-phosphoethanolamine (Avanti Polar Lipids). First, the amino-group was protected with the *tert*-butoxycarbonyl protecting group (di-*tert*-butyldicarbonate/triethylamine; 98% overall yield). The N-protected lyso-PE obtained was coupled with pacFA using EDCI/DMAP activation in a good yield (52%). The final deprotection step was achieved with TFAA to generate pacPE (35%, overall yield).

**Antibodies**. Antibodies used were mouse monoclonal anti-β-actin (Sigma-Aldrich, A1978; IB 1:50,000), rabbit polyclonal anti-FLAG (Cell Signaling, 2368; IB 1:1,000), mouse monoclonal anti-mitochondrial surface protein p60 (Millipore, MAB1273; IB 1:1,000), rabbit polyclonal anti-VDAC1 (Cell Signaling, 4661; IB 1:1,000), goat polyclonal anti-VDAC2 (Abcam, Ab37985; IB 1:1,000), rabbit polyclonal anti-VDAC3 (Abcam, Ab80452; IB 1:1,000), mouse monoclonal anti-TOM20 (Millipore, Mabt166; IF 1:200), rabbit polyclonal anti-TOM40 (Abcam, Ab185543; IB 1:1,000), rabbit polyclonal anti-HA (Invitrogen, 715500; IB 1:1,000; IF 1:200), rat monoclonal anti-HA (Roche, 12158167001; IB 1:1,000), rabbit monoclonal anti-Bax (Cell Signaling, 5023; IB 1:1,000), rabbit polyclonal anti-cleaved caspase-3 (Cell Signaling, 96611; IB 1:1,000), mouse monoclonal anti-PARP-1 (Santa Cruz, sc8007; IB 1:1,000) and rabbit polyclonal anti-calnexin (Santa Cruz, sc11397; IB 1:1,000). Goat anti-mouse (31430; IB 1:5,000), goat anti-rabbit (31460; IB 1:5,000) and donkey anti-goat IgG conjugated to horseradish peroxidase (pa1-28664; IB 1:5,000) were from Thermo Fischer Scientific. Cy™-dye-conjugated donkey anti-mouse and donkey anti-rabbit antibodies (715-225-150, 715-225-152, 715-165-150, 715-165-152, 715-175-150 and 715-175-152; IF 1:250 each) were from Jackson ImmunoResearch Laboratories.

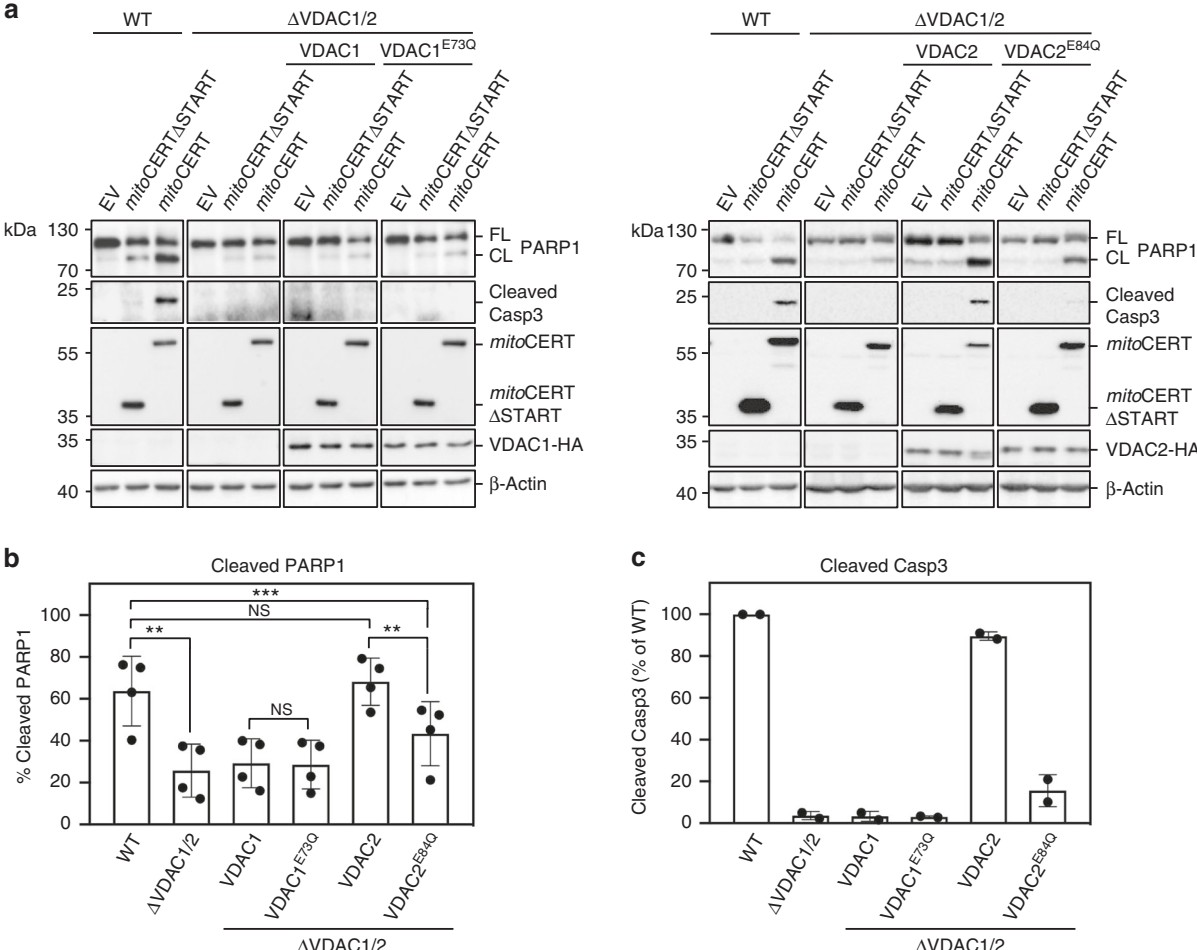

**Fig. 7** Ceramide-induced apoptosis critically relies on Glu84 in VDAC2. **a** WT and ΔVDAC1/2 HCT116 cells stably transduced with HA-tagged VDAC1, VDAC1$^{E73Q}$, VDAC2, or VDAC2$^{E84Q}$ were transfected with empty vector (EV), Flag-tagged mitoCERT, or Flag-tagged mitoCERTΔSTART. At 24 h post transfection, cells were processed for immunoblotting with antibodies against PARP1, cleaved caspase-3 (Casp3), the Flag-epitope, the HA-epitope, and β-actin. FL full-length, CL cleaved. **b** Quantitative analysis of PARP1 cleavage in cells treated as in **a**. Data are means ± s.d.; $n = 4$; $*p < 0.05$, $**p < 0.01$ and $***p < 0.001$ by two-tailed paired $t$-test. **c** Quantitative analysis of cleaved Casp3 in cells treated as in **a**. Data are means ± s.e.; $n = 2$. Source data

**DNA constructs**. For expression of human VDACs in *E. coli*, the corresponding cDNAs were PCR amplified using Phusion high-fidelity DNA polymerase (Thermo Fischer Scientific) and inserted via *Nde*I and *Xba*I (VDAC1) or *Xho*I and *Xba*I sites (VDAC2) into bacterial expression vector pCold I (Takara Bio, USA). For retroviral transduction studies, DNA fragments encoding human VDACs with a C-terminal HA tag (YPYDVPDYA) were created by PCR and inserted via *Not*I and *Xho*I sites into lentiviral expression vector pLNCX2 (Takara Bio, USA). Single amino acid substitutions were introduced using the QuikChange II site-directed mutagenesis method (Alignet, USA). Primers used for cloning and site-directed mutagenesis are listed in Supplementary Table 3. Mammalian expression constructs encoding FLAG-tagged mitoCERT and mitoCERTΔSTART were described previously[25]. All expression constructs were verified by DNA sequencing.

**Cell culture and transfection and RNAi**. Human cervical carcinoma HeLa cells (ATCC CCL-2) were cultured in Dulbecco's modified Eagle's medium (DMEM) supplemented with 4.5 g/l glucose, 2 mM L-glutamine and 10% FBS. Human colon carcinoma HCT116 cells (ATCC CCL-247) were cultured in McCoy's medium supplemented with 10% FBS. Human embryonic kidney HEK293T cells (ATCC CRL-3216) were cultured in DMEM supplemented with 10% FBS. Cells were transfected with DNA constructs using Effectene (Qiagen) according to the manufacturer's instructions unless stated otherwise. Treatment of HeLa cells with siRNA (Qiagen) was performed using Oligofectamine reagent (Invitrogen) according to the manufacturer's instructions. siRNA target sequences were: NS (nonsense), 5′-AAUUCUCCGAACGUGUCACGU-3′; VDAC1, 5′-ACACUA GGCACCGAGAUUAUU-′3; VDAC2, 5′-AAUACAAGUGGUGUGAGUAUU-3′.

**Generation of VDAC KO cell lines**. To knock out VDAC1 and VDAC2 in HCT116 cells, we obtained a mix of three different CRISPR/Cas9 plasmids per gene

and the corresponding HDR plasmids from Santa Cruz (sc-418200, sc-416966). The VDAC1-specific gRNA sequences were: A/sense, 5′-TTGAAGGAATTTACA AGCTC-3′; B/sense, 5′-CGAATCCATGTCGCAGCCC-3′; C/sense, 5′-CTTACA CATTAGTGTGAAGC-3′. The VDAC2-specific gRNA sequences were: A/sense, 5′-AGAAATCGCAATTGAAGACC-3′; B/sense, 5′-GCCCTTAAGCAGCACAG CAT-3′; C/sense, 5′-TAATGTGACTCTCAAGTCCT-3′. HCT116 cells were transfected with both plasmid mixes and grown for 48 h without selection. Next, cells were grown for 2 weeks under selective pressure with 2 µg/ml puromycin. Individual drug-resistant clones were picked and analyzed for VDAC1 and VDAC2 expression by immunoblot analysis. A VDAC1/2 double KO cell line was generated from ΔVDAC1 cells as described above following ejection of the puromycin selectable marker using Cre vector (Santa Cruz, sc-418923) according to the manufacturer's instructions.

**Retroviral transduction**. HCT116 VDAC1/2 double KO cells stably expressing HA-tagged VDAC1, VDAC1$^{E73Q}$, VDAC2, or VDAC2$^{E84Q}$ were created by retroviral transduction. To this end, HEK293T cells were co-transfected with pLNX2-VDAC-HA expression constructs and packaging vectors (Clontech) using Lipofectamine 3000 (Invitrogen) according to the manufacturer's instructions. The culture medium was changed 6 h post transfection. After 48 h, the retrovirus-containing medium was harvested, filtered through a 0.45 µm filter, mixed 1:1 (v/v) with McCoy's growth medium, supplemented with 8 µg/ml polybrene, and used to transduce HCT116 VDAC1/2 double KO cells. Hygromycin (300 µg/ml) was added 6 h post-infection and selective medium was exchanged daily. After 3–5 days, positively transduced cells were selected and analyzed for expression of HA-tagged VDACs by immunoblot analysis and immunofluorescence microscopy.

**Reconstitution of recombinant VDACs**. VDAC-encoding pCold I constructs were transformed in *E. coli* BL21 (DE3) ΔOmp9 cells (a kind gift from Dr. Lars-Oliver Essen, Philips-Universität Marburg, Germany). Transformants were grown at 37 °C to early exponential phase in LB medium containing 100 µg/ml ampicillin and cooled for 30 min at 4 °C prior to addition of 1 mM IPTG. Growth was continued for 24 h at 15 °C. Cells were collected by centrifugation and lysed in TEN buffer (50 mM Tris/HCl pH 8.0, 100 mM NaCl) supplemented with 2.5% Triton X-100 and protease inhibitor cocktail (PIC; 1 µg/ml apoprotinin, 1 µg/ml leupeptin, 1 µg/ml pepstatin, 5 µg/ml antipain, 157 µg/ml benzamide) through micro-tip sonication. Inclusion bodies were collected by centrifugation (30 min, 2.500 × $g$, 4 °C), washed three times in TEN buffer with, and then three times in TEN buffer without, Triton X-100. Inclusion bodies were diluted 1:10 by dropwise addition into 25 mM Na$^+$PO$_4$ pH 7.0, 100 mM NaCl, 6 M guanidine hydrochloride, 1 mM EDTA and 10 mM DTT while stirring, and stirred overnight at 4 °C. Next, the suspension was diluted 1:10 by dropwise addition to 25 mM Na$^+$PO$_4$ pH 7.0, 100 mM NaCl, 1 mM EDTA, and 2.2% lauryldimethylamine oxide (LDAO), and stirred overnight at 4 °C. Finally, the suspension was diluted 1:10 by dropwise addition to 25 mM Na$^+$PO$_4$ pH 7.0, 10 mM NaCl, 1 mM EDTA, 0.1 % LDAO, and 1 mM DTT, stirred overnight at 4 °C, and loaded on a Fractogel EMD-SE Hicap cation-exchange column (Merck Millipore). VDAC proteins were eluted with a linear NaCl concentration gradient in 25 mM Na$^+$PO$_4$ pH 7.0, 10 mM NaCl, 1 mM EDTA, 1 mM DTT, and 0.1% LDAO on a ÄKTAprime plus protein purification system (GE Healthcare Life Sciences). Peak fractions were pooled, concentrated on an Amicon Ultra-4 unit (MWCO 10 kDa; Merck Millipore), and loaded on a Superose 12 10/300 GL size exclusion column (GE Healthcare Life Sciences). Elution was in 10 mM Tris/HCl, pH 7.0, 100 mM NaCl, and 0.05% LDAO. Purified VDAC proteins in peak fractions were pooled, quantified using Amido Black[58], and then reconstituted at a concentration of 1 mg/ml in egg PC vesicles supplemented with 1 mol% NBD-PC at a protein to lipid ratio of 1:300 (mol/mol). To this end, egg PC and NBD-PC dissolved in CH$_3$Cl were dried and dissolved in buffer R (100 mM KCl, 10 mM MOPS/Tris pH 7.0) by vortexing and sonication. LDAO was added in 10-fold molar excess over lipids. The concentration of LDAO in the protein sample was adjusted to that in the lipid sample. Both samples were incubated separately at RT for 20 min, mixed at a 1:1 (v/v) ratio, and incubated again for 30 min. SM2 Biobeads (Bio-Rad Laboratories) pretreated according to the manufacturer's instructions were added to the lipid–protein mixture in 30-fold excess over detergent (w/w). After incubation overnight at 4 °C on a rotating wheel, the beads were removed by centrifugation and the proteoliposome-containing supernatant was aliquoted, snap-frozen in liquid N$_2$, and stored at −80 °C. To check reconstitution efficiency, an aliquot of the proteoliposomes was subjected to density flotation analysis. To this end, proteoliposomes were mixed 1:1 (v/v) with 80% Accudenz (Accurate Chemical & Scientific Corporation, USA) in buffer R, transferred to the bottom of a 5 ml ultracentrifuge tube, and overlayed with 30%, 20%, 10%, and 0% Accudenz prepared in reconstitution buffer. After centrifugation for 1 h at 100,000 × $g$ at 4 °C, 10 × 0.5 ml fractions were collected from top to bottom. Fractionation profiles of NBD-PC and VDACs were determined by TLC analysis and SDS-PAGE followed by Coomassie blue staining, respectively.

**Photoaffinity labeling of recombinant VDACs**. Liposomes used for photoaffinity labeling of reconstituted VDACs were prepared from a defined lipid mixture (DOPC/DOPE/pacLipid, 80/20/1 mol%) in CHCl$_3$/methanol (9/1, v/v). In brief, 10 µmol of total lipid was dried in a Rotavap and the resulting lipid film was resuspended in 1 ml buffer L (50 mM Tris-HCl pH 7.4, 50 mM NaCl) by vigorous vortexing and sonication, yielding a 10 mM lipid suspension. Liposomes with an average diameter of ~100 nm were obtained by sequential extrusion of the lipid suspension through 0.4, 0.2, and 0.1 µm track-etched polycarbonate membranes (Whatman-Nucleopore) using a mini-extruder (Avanti Polar Lipids). VDAC proteoliposomes were diluted in buffer R to a final protein concentration of 0.1 mg/ml, mixed with an equal volume of 10 mM liposomes containing 1 mol% pacLipid, and incubated for 30 min at 37 °C with gentle shaking. For the experiments shown in Figs. 3 and 4, VDAC proteoliposomes were incubated with pacLipids added from ethanolic stocks (0.2 nmol in 100 µl reaction volume, unless indicated otherwise). To determine the impact of pH on pacCer labeling of VDAC1, VDAC1 proteoliposomes were collected by high-speed centrifugation and then resuspended in the following buffers prior to photolabeling: (a) 100 mKCl, 50 mM MES-NaOH pH 5.0; (b) 100 mKCl, 50 mM MES-NaOH pH 6.0; (c) 100 mKCl, 10 mM MOPS/Tris pH 7.0; (d) 100 mKCl, 50 mM HEPES pH 8.0. The mixtures were placed on ice and irradiated for 90 s using a 1000 W mercury lamp equipped with a dichroic mirror and a 345 nm bandpass filter (Newport) at 30 cm distance. Proteins were recovered by chloroform–methanol precipitation after external addition of BSA as a carrier and the air-dried protein pellet was dissolved in 1% SDS in PBS with vigorous shaking for 10 min at 70 °C. Samples were click-reacted with 80 µM AF647-N$_3$ in 1 mM TCEP (Tris(2-carboxyethyl)phosphine hydrochloride), 0.1 mM TBTA (Tris[(1-benzyl-1H-1,2,3-triazol-4-yl)methyl] amine), and 1 mM CuSO$_4$ for 1 h at 37 °C. After addition of 0.25 vol of 5× Sample buffer (0.3 M Tris/HCl, pH 6.8, 10% SDS, 50% glycerol, 0.025% bromphenol blue, and 10% β-mercaptoethanol), samples were boiled for 5 min at 95 °C, subjected to SDS-PAGE and analyzed by IGF using a Typhoon FLA 9500 (GE Healthcare) with a 635 nm laser and LPR filter. Fluorescence intensities were quantified using ImageQuant TL software. Next, gels were stained with Coomassie blue and the amount of recombinant

protein was determined by measuring the staining intensity using Image Lab 5.2 software (Bio-Rad Laboratories). Fluorescence intensities of recombinant protein were corrected for background fluorescence in the same lane and then divided by the total amount of protein. The specific fluorescence intensity of UV-irradiated protein was determined after subtraction of fluorescence intensity of non-UV-irradiated protein.

**Identification of pacCer-photolabeled proteins**. Post-nuclear supernatants, cytosol, total membranes, and membrane fractions enriched in mitochondria or ER were prepared from HeLa cells[25]. Subcellular fractions were diluted in buffer R (10 mM Tris, pH 7.4, 0.25 M sucrose) supplemented with 0.1 mM PMSF and PIC to 0.45–0.70 mg/ml total protein and then mixed with an equal volume of 10 mM liposomes containing 1 mol% pacCer prepared in buffer R as above. Samples were incubated for 60 min at 37 °C with gentle shaking, placed on ice, and then UV irradiated for 90 s. Proteins were recovered by chloroform–methanol precipitation, click reacted with AF647-N$_3$ and subjected to IGF as above. For identification of the mitochondria-associated 33kDa-protein band, an HeLa cell-derived membrane fraction enriched in mitochondria (20 µg total protein) was photolabeled with pacCer as above. Experiments in which pacCer was omitted from the liposomes served as controls. Proteins were recovered by chloroform–methanol precipitation, dissolved in 50 µl 1% SDS PBS by vigorous vortexing and sonication in a water bath. After addition of 50 µl PBS, the sample was click-reacted with 1 mM TAMRA-Biotin-N$_3$ as above. Next, two-third of the sample was mixed with 100 µl slurry of NeutrAvidin™ Agarose Resin (Thermo Fisher Scientific, USA) pre-equilibrated in PBS/0.5% SDS and incubated for 2 h at RT while shaking. The beads were washed five times in 0.6 ml PBS/0.5% SDS and then incubated in Sample buffer for 5 min at 95 °C to elute bound proteins. Proteins in total input, flow-through, wash, and eluate were recovered by chloroform–methanol precipitation and processed for IGF analysis as above. The pacCer-labeled 33-kDa-protein band in the eluate fraction was excised from the gel and digested by trypsin (Promega, V5111) for 6 h at 37 °C. Protein fragments were collected, concentrated, and analyzed by LC-MS/MS in an amaZon Spped ETD ion Trap mass spectrometer (Bruker, USA) connected to an UltiMate 3000 nano LC system (Thermo Fisher Scientific). Peptides were separated on a C18 column using a two-buffer system with water/acetonitrile/formic acid (99/1/0.1, v/v/v) and water/acetonitrile/formic acid (20/80/0.1, v/v/v). Raw MS data were converted to peak lists using Data Analyzer 4.1 (Bruker, USA) and noise filtered using an in-house developed script. The spectra were searched with Mascot (precursor mass tolerance 0.8 Da; product mass tolerance 0.4 Da; 2 missed cleavages) against all human proteins in the Swissprot (v56.2) database. Peptide identifications were accepted with a score greater than 20 and a *p*-value smaller than 0.01, and proteins were identified with at least two unique peptides.

**MD simulations**. For VDAC MD simulations, a well-resolved structure of mouse VDAC1 (PDB: 4C69 at 2.8 Å resolution)[41] was selected as a main template. This structure shares a β-barrel backbone RMSD of 1.7 Å with a structure of VDAC2 from zebrafish (PDB: 4BUM)[42]. Based on the assumption of identical secondary structures, VDAC1 side chains were mutated to the mouse sequences of VDAC2 (NCBI ID:NP_035825.1) and VDAC3 (NCBI ID:NP_035826.1) to obtain structures of all three isoforms for comparison using the PyMOL software. The mutation process was semi-automated and leveraged the used CG representation (see below), in which side chains are represented by up to four particles and are therefore simple to replace. The N-terminal α-helix of the VDAC2 was truncated to match VDAC1 sequence length (with no impact to lipid interactions because the helix sits inside the channel's β-barrel). VDACs were embedded in an outer mitochondrial membrane mimic with about 630 lipids using the insane script[59]. Membrane composition was based on Horvath and Daum[43], with a mixture of POPC/POPE/POPI/cholesterol (52/14/19/15, mol%) in the inner leaflet and POPC/POPE/POPI/cholesterol (42.5/32/5/15.5, mol%) supplemented with 5 mol% C$_{16}$-ceramide in the outer leaflet. About 150 mM NaCl was added to the system, plus an excess of Na$^+$ ions to reach charge neutrality. The Martini coarse-grain (CG) force field[60] was used to model the simulated systems, together with the ElNeDyn elastic-network approach to restrain the protein secondary structure[61]. Simulations were run with GROMACS version 5.0[62] in the isothermal–isobaric (NpT) ensemble, at 300 K and 1 bar. Temperature was controlled using the v-rescale thermostat with a coupling constant of 1.0 ps. Pressure was coupled semi-isotropically, independently in $xy$ and $z$, using the Parrinello–Rahman barostat with a coupling constant of 12.0 ps and a compressibility of $4.5 \times 10^{-5}$ bar$^{-1}$. Standard Martini parameters were used to represent interparticle interactions[60,63]: electrostatic interactions were modeled with a Coulombic potential cutoff at 1.2 nm, shifted to zero along the entire range, in conjunction with an implicit screening constant of 15. The Van der Waals interactions were calculated using a shifted Lennard–Jones potential, with a cutoff of 1.2 nm and a shift to zero from 0.9 nm. A simulation integration time step of 20 fs was used, and particle neighbor-searching was performed over a 1.4 nm radius once every 10 simulation steps. As the apparent pKa of the membrane-facing glutamate E73 in VDAC1 is closely tuned to the physiological pH of the cytosol (pKa ~7.4)[51], main simulations of VDAC1 and VDAC2 were initially done with deprotonated E73/E84 in six replicates (0.3 ms total) each. Other simulated systems include VDACs with protonated E73/E84, or with E73Q/E84Q or E73D/E84D mutations. VDAC1 with deprotonated E73 was also simulated in 100% DMPC. The total simulation time for all

systems was 1.23 ms (Supplementary Table 2). Trajectories were saved and analyzed every 600 ps.

**Simulation analysis**. Two main types of analysis were performed: contact analysis and space occupation analysis. Contact analyses were performed either as residue-discriminated analyses or as residence-time distributions. Residue-discriminated contacts were calculated by finding all frames of ceramide (particles AM1 or AM2) or cholesterol (particle ROH) head groups within 7 Å of any relevant residue particle. To allow comparison between simulations of different lengths, the number of in-contact frames was normalized by the total number of frames. Contact residence-time distributions were plotted by logarithmically histogramming the duration of all individual continuous binding events of each single ligand to the VDAC-binding site. To correctly convey the impact of long-term binding events, $y$-values indicate the sum of binding durations in that bin. To compare between simulations of different duration, the $y$-values were further normalized for the respective total simulation time. Analysis of continuous contact times is very sensitive to the use of a sharp cutoff, and long binding events will be underestimated if the ligand briefly drifts past the threshold, even if not actually leaving the binding site. To correct for this, for binding duration analysis a contact was defined when a head group particle (AM1 or AM2 in ceramide; PO4 or NC3 in DMPC) came within 8 Å of at least two groups of residues, out of three defined on the β-strands that compose the binding site (VDAC1 residues 58–60, 73–75, and 81–83). Spuriously brief contacts, or spuriously brief losses of contact, were further filtered by applying a smoothing window five-frames wide; if there were contacts on three or more of those frames the center frame was set to be in-contact, otherwise it was set to be not-in-contact. Binding events that were still bound at the end of a simulation run were disregarded. Space occupation analyses were performed by gridding the simulation box at 5 Å spacing and comparing ceramide versus cholesterol occupancy. The number of occupied frames for each cell was calculated for the entire simulation time, and then normalized by the total number of frames, yielding a grid of occupancy values ranging from zero (never present in that cell) to one (always present). Occupancy volumes were drawn as iso-occupancy surfaces at given thresholds (10% for ceramide, 20% for cholesterol), adjusted to compensate for the difference in the number of lipid molecules. To allow meaningful occupancy surfaces to be drawn relative to the protein, its position and rotation in the $xy$ plane were aligned beforehand across all analyzed trajectories.

**Algorithms and statistical analysis**. Analysis algorithms were programmed in Python, using MDAnalysis (https://www.mdanalysis.org)[64,65] and NumPy[66] packages. The MDreader package (https://github.com/mnmelo/MDreader) was used to allow analysis parallelization, which was essential in tackling the multi-gigabyte trajectory analysis in a short time. Two types of statistical analyses were performed on the distribution of contact residence times: data are presented as time-weighted average values, with 95% confidence intervals estimated by a 10k-resample bootstrapping of the time-weighted averaging, and the significance level for the difference between binding durations was obtained by a continuity-corrected one-tailed Mann–Whitney–Wilcoxon $U$ test. The Visual Molecular Dynamics (VMD) software[67] was used to create images and movies.

**Reporting summary**. Further information on experimental design is available in the Nature Research Reporting Summary linked to this article.

## Data availability

Data supporting the findings of this manuscript are available from the corresponding authors upon reasonable request. A Reporting Summary for this Article is available as a Supplementary Information file. Full scans of gels and blots are provided in Supplementary Information. The source data underlying Figs. 3d, 4b, 5b, 6c, 7b, 7c and Supplementary Table 1 are provided as a Source Data file.

## Code availability

Trajectory analysis programs, as well as protein structures and models, are available from the download section of the Melo lab's website, at http://www.itqb.unl.pt/labs/multiscale-modeling/.

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

## Acknowledgements

This work was supported by the Deutsche Forschungsgemeinschaft (Projects SFB944-P14 and HO3539/1-1) to J.C.M.H., a Marie Curie Intra-European Fellowship to J.G.M.M. (Project 289278), and a German Egyptian Research Long-term Scholarship to D.G.H. (Project 57222240). M.N.M was supported by Project LISBOA-01-0145-FEDER-007660 (Microbiologia Molecular, Estrutural e Celular) funded by FEDER funds through COMPETE2020—Programa Operacional Competitividade e Internacionalização (POCI) and by national funds through FCT—Fundação para a Ciência e a Tecnologia. M.N.M. and G.R. acknowledge the National Laboratory for Scientific Computing (LNCC/MCTI, Brazil) for providing HPC resources of the SDumont supercomputer (http://sdumont.lncc.br). Additional support was provided by the NIH (Project 1R21AI124225-01A1) to F.G.T.

## Author contributions

J.C.M.H. conceptualized the study and wrote the manuscript, with critical input from S.D., S.B., J.G.M.M., and M.N.M. J.G.M.M. designed and performed the photolabeling, fractionation, purification, and identification of ceramide-binding proteins. M.N.M. designed, performed, and analyzed the computer simulation studies, with critical input from G.R. and S.J.M. S.K. synthesized photoactive and clickable lipid analogs. S.B., D.G.H., H.J., D.M., and M.S. designed and performed the reconstitution and photo-labeling of recombinant protein channels. S.D. and P.N. created and characterized mutant cell lines, with critical input from F.G.T. S.D. designed and performed transfection studies and photolabeling of purified mitochondria, with critical input from D.G.H.

## Additional information

**Competing interests:** The authors declare no competing interests.

