## [Peer Review File · Nature Communications]

Reviewers' Comments:

Reviewer #2:

Remarks to the Author:

The authors identify VDAC 1 and VDAC 2 as potential ceramide binding proteins using photolabeling and in gel fluorescence. Mass spectrometry, siRNA, and isoform specific antibodies confirm that VDAC 1 and VDAC 2 are photolabeled by pacCer (ceramide photolabeling reagent) in mitochondria. Excess C16-Ceramide is shown to reduce pacCer photolabeling in VDAC1, consistent with VDAC1 containing a specific ceramide binding site. Molecular modeling is used to postulate that ceramide binds to E73 on VDAC1 and E84 on VDAC2, consistent with their find that E73Q and E84Q markedly reduce VDAC photolabeling with pacCer. The authors then show that VDAC 2 is required for ceramide-induced apoptosis and that VDAC 2 E84Q reduces ceramide-induced apoptosis. The findings in this manuscript are of potential biological importance and could have interest to a broad audience. There are some aspects of the manuscript, particularly the identification of a specific ceramide binding site, that are inadequately demonstrated, and should be supported by additional experimental evidence.

Ceramide labels VDAC1 E73Q and VDAC2 E84Q:

The authors provide strong evidence that there is specific binding of ceramide to VDAC 1, including photolabeling data, molecular simulation studies and prevention of photolabeling by mutation of E73 and by competition with excess ceramide. The identification of E73 as a specific photolabeling site is indirect and potentially inconsistent with the simulation data. The photo-crosslinking group in pacCer is an aliphatic diazirine located on the acyl group, not on the ceramide head group. If the charged ceramide head group binds to the E73 side chain (as indicated in coarse grain simulation), E73 should not be labeled by an acyl chain labeling group. Aliphatic diazirines preferentially label nucleophilic amino acids, particularly acidic amino acids, suggesting that pacCer may be diffusing to this preferred labeling site rather than directly binding to it. It is also not certain that pacCer is labeling E73; the E73 residue has been shown to be functionally important and the E73Q mutation could result in a conformation that is not labeled by pacCer. Two experiments could greatly strengthen this aspect of the manuscript: (1) the lifetime of photoactivated pacCer should be assessed to assure that it is not diffusing to a preferred labeling residue after UV irradiation; (2) the pacCer photo-labeled residue should be specifically identified by endoproteolytic digestion of pacCer-labeled VDAC and mass spectrometric sequencing. The manuscript proposes that ceramide binding to VDAC2 is important for ceramide-induced apoptosis. The paper would therefore also be strengthened by showing C16-Ceramide reduces pacCer labeling of VDAC2, not just VDAC1. Similarly, the labeling of E84 should be directly determined by sequencing rather than indirectly by modeling and mutagenesis.

Ceramide binding site is specific and not also occupied by PC or cholesterol:

The authors assert that the proposed binding sites on VDAC1 and VDAC2 are specific to ceramide and use molecular docking studies as evidence. However, pacCer, pacPC and pacChol all photolabel VDAC 1, with labeling largely or partially prevented by the E73Q mutation. Similar data are shown for VDAC 2, with the apparent exception that E84Q does not prevent pacChol labeling. (Of note, there is a large error bar for the VDAC 2 pacChol data with an n=2, suggesting two very discrepant data points. All of the data points should be shown, rather than the mean and n should be increased). Additionally, the structures of all of the photolabeling analogues should be shown; knowledge of where the photoactivatable group is positioned is essential to interpreting the specificity of labeling.

There is experimental evidence for specific cholesterol binding to a site containing E73 on VDAC 1 (ref #21, Budelier et. al. 2017). While this paper is referenced to make a different point, cholesterol binding is not mentioned. The referenced paper maps three residues for a cholesterol binding pocket containing E73 on VDAC1 and shows that cholesterol binds to that pocket in both

WT VDAC1 and E73Q VDAC1. The authors should reference the paper in context and consider the possibility that multiple lipids are binding to the same site. To ascertain whether cholesterol and ceramide bind to the same site it would be useful to determine if excess cholesterol prevents pacCer labeling of VDAC1/2.

Ceramide inhibition of apoptosis is dependent on VDAC2:

The data in Figure 4 show a clear role for VDAC2 in ceramide-induced apoptosis. The data in Figure 4 and supplementary Figure 8 also support that VDAC2 E84Q reduces ceramide-induced, VDAC2-dependent apoptosis. This represents the most important point of the paper (a specific ceramide binding site on VDAC2 is required for ceramide-induced apoptosis), but was only done in duplicate. The results show a small effect of E84Q on PARP1 cleavage and a more substantial effect on Casp3. These results (figure 8) would benefit from more replicates (with $n \geq 3$), error bars, quantitation and statistical analysis.

Minor points:

1. In the molecular docking studies, why is the threshold for cholesterol set at 20% occupancy while the threshold for ceramide is set at 10% (figure 2d)?

Reviewer #3:

Remarks to the Author:

The authors combine coarse-grained computer simulations with functional measurements to show that the mitochondrial anion channel VDAC2 bind ceramide through a membrane-embedded glutamate residue. This finding suggests a role for VDAC2 in ceramide translocation through the membrane and subsequent apoptosis.

I primarily reviewed the computational aspects of the paper. I like the synergistic use of simulations and experiments, which reveals interesting molecular details such as the likely ceramide binding sites. I have a number of mostly technical comments that should be clarified.

1) How was the VDAC2 model created? What is the sequence identity between the template mouse VDAC1 and VDAC2? If the sequence identity is lower than 90% or so then a tool such as MODELLER should have been used.

2) How reliable are Coulombic interactions in CG FF such as Martini? This is important for assessing the interaction between E73 and ceramide and other lipids.

3) On line 81 "fine grain simulation analysis" is mentioned. "Fine grain" is jargon and it is not quite clear what the authors mean --- atomic resolution simulations?

4) What is the protonation state of E73 and how was it determined? What is the pKa of the residue? Do the results depend on it being protonated/deprotonated?

5) Does a E73D mutant retain ceramide binding?

Methods

6) Is epsilon=15 standard for Martini? Given the importance of electrostatics in the current study,

it should be clearly discussed what the known limitations of the CG FF are in this regard.

7) MDAnalysis is not cited (only website given)

RESPONSE TO REVIEWERS' COMMENTS

Reviewer #2

The authors identify VDAC1 and VDAC2 as potential ceramide binding proteins using photolabeling and in gel fluorescence. Mass spectrometry, siRNA, and isoform specific antibodies confirm that VDAC1 and VDAC2 are photolabeled by pacCer (ceramide photolabeling reagent) in mitochondria. Excess C16-Ceramide is shown to reduce pacCer photolabeling in VDAC1, consistent with VDAC1 containing a specific ceramide binding site. Molecular modeling is used to postulate that ceramide binds to E73 on VDAC1 and E84 on VDAC2, consistent with their find that E73Q and E84Q markedly reduce VDAC photolabeling with pacCer. The authors then show that VDAC2 is required for ceramide-induced apoptosis and that VDAC2 E84Q reduces ceramide-induced apoptosis. The findings in this manuscript are of potential biological importance and could have interest to a broad audience. There are some aspects of the manuscript, particularly the identification of a specific ceramide binding site, that are inadequately demonstrated, and should be supported by additional experimental evidence.

1) Ceramide labels VDAC1 E73Q and VDAC2 E84Q: The authors provide strong evidence that there is specific binding of ceramide to VDAC1, including photolabeling data, molecular simulation studies and prevention of photolabeling by mutation of E73 and by competition with excess ceramide. The identification of E73 as a specific photolabeling site is indirect and potentially inconsistent with the simulation data. The photo-crosslinking group in pacCer is an aliphatic diazirine located on the acyl group, not on the ceramide head group. If the charged ceramide head group binds to the E73 side chain (as indicated in coarse grain simulation), E73 should not be labeled by an acyl chain labeling group. Aliphatic diazirines preferentially label nucleophilic amino acids, particularly acidic amino acids, suggesting that pacCer may be diffusing to this preferred labeling site rather than directly binding to it. It is also not certain that pacCer is labeling E73; the E73 residue has been shown to be functionally important and the E73Q mutation could result in a conformation that is not labeled by pacCer. Two experiments could greatly strengthen this aspect of the manuscript: (1) the lifetime of photoactivated pacCer should be assessed to assure that it is not diffusing to a preferred labeling residue after UV irradiation; (2) the pacCer photo-labeled residue should be specifically identified by endoproteolytic digestion of pacCer-labeled VDAC and mass spectrometric sequencing.

We appreciate the reviewer's thoughtful comments but want to emphasize that our manuscript does not claim at any point that the membrane-exposed Glu in VDAC1/2 acts as a specific photolabeling site for pacCer. We acknowledge that our photolabeling experiments on wild-type and mutant channels provide only indirect evidence for the binding of the ceramide head group to the Glu residue as observed in the simulations. However, identification of the pacCer photo-labeled residue by endoproteolytic digestion of pacCer-labeled VDAC and mass spectrometric sequencing, while technically feasible, will unlikely yield a meaningful outcome. Such approach would require application of excess pacCer on detergent-solubilized VDACS, conditions that deviate dramatically from both simulations and photo-labeling experiments on VDACS embedded in intact bilayers that contain only a minor fraction of pacCer (≤ 1 mol% of total lipid). Consequently, identification of the membrane-exposed Glu or a neighboring residue as the site of photo-labeling would not enable one to firmly validate or disprove the transient contacts between the ceramide head group and Glu residue observed in the simulations. Also, introducing the photo-crosslinkable diazirine in the ceramide head would not be very meaningful, as this would likely perturb such contacts.

While we cannot fully exclude the possibility that mutation of the membrane-exposed Glu in VDAC1/2 results in a conformation that is not labeled by pacCer, this appears unlikely. The structure of an E73V hVDAC1 mutant has been solved (Jaremko et al., 2016, Angewandte 55, 10518-10521; doi: 10.1002/anie.201602639). In that work, in which the mutation is chemically farther from wild-type than E73Q, the beta-barrel structure was clearly retained, with only a minor elliptic distortion compared to mVDAC1. This supports our assumption that the substitution of a Gln should not affect the structure of VDAC locally to the binding site to the point of lowering

ceramide binding on its own. Moreover, in response to comment #4 of Reviewer 3, we performed fresh MS simulations indicating that the protonation state of the Glu residue is a critical determinant of ceramide binding (new Figs. 3a, b). Consistent with this finding, we also show that a drop in pH is accompanied by a reduced photolabeling of VDAC1 by pacCer (new Figs. 3c, d). These results are now described on p. 5 (lines 4-12) and p. 6 (lines 13-15), and their functional implications discussed on p. 9 (lines 2-9 & 20-27) and p. 10 (lines 1-3).

2) The manuscript proposes that ceramide binding to VDAC2 is important for ceramide-induced apoptosis. The paper would therefore also be strengthened by showing C16-Ceramide reduces pacCer labeling of VDAC2, not just VDAC1. Similarly, the labeling of E84 should be directly determined by sequencing rather than indirectly by modeling and mutagenesis.

As requested, we now performed competition experiments showing that excess C16-ceramide also reduces pacCer labeling of VDAC2, not just VDAC1 (new Fig. 5b). For VDAC2, a slightly higher concentration of C16-ceramide was required to obtain a significant reduction in labeling. Regarding the photo-labeling of E84 in VDAC2, please see our reply to comment #1.

3) Ceramide binding site is specific and not also occupied by PC or cholesterol: The authors assert that the proposed binding sites on VDAC1 and VDAC2 are specific to ceramide and use molecular docking studies as evidence. However, pacCer, pacPC and pacChol all photolabel VDAC1, with labeling largely or partially prevented by the E73Q mutation. Similar data are shown for VDAC2, with the apparent exception that E84Q does not prevent pacChol labeling. (Of note, there is a large error bar for the VDAC2 pacChol data with an n=2, suggesting two very discrepant data points. All of the data points should be shown, rather than the mean and n should be increased).

We do not assert that the proposed binding site on VDAC1/2 can be exclusively occupied by ceramide and do not rule out that the same site binds lipids other than ceramide. In fact, our coarse grain simulations (Supplementary Fig. 4a,b) and a previous atomic resolution simulation analysis (Villenger et al., 2010, PNAS 107, 22546-22551; Ref. 45) revealed occasional contacts between the PC head group and the membrane-facing Glu residue. However, these encounters are much shorter-lived (≤ 5 ns; Supplementary Fig. 4b) and extremely rare in comparison to those involving ceramide, which displayed average residence times of 0.8 μ s and 1.2 μ s for VDAC1 and VDAC2, respectively (Supplementary Fig. 4c; see also p. 5, lines 19-24). As requested, we now performed another round of photolabeling experiments to increase the total number of independent measurements for each pacLipid and included all data points in the bar graphs (new Fig. 4b). We acknowledge that there is considerable variability in the data, notably regarding the pacPC labelling of wild-type VDAC1/2 and pacDAG/pacChol labelling of wild-type VDAC2. However, this may not be that surprising as each round of experiments was performed with distinct preparations of proteoliposomes and pacLipids. Nevertheless, we found that substituting Gln for the membrane-exposed Glu in VDAC1/2 resulted mostly in no or only a slight reduction of pacChol labelling (new Figs. 4b, 5); this is in contrast to its major impact on pacCer and pacPC labelling.

4) Additionally, the structures of all of the photolabeling analogues should be shown; knowledge of where the photoactivatable group is positioned is essential to interpreting the specificity of labeling.

As requested, we now included the structures of all pacLipids in new Supplementary Fig. 6. As shown in Fig. 4, pacCer and pacPC readily label VDAC1/2 whereas pacPE and pacDAG do not, in spite of carrying the diazirine at the same position. Consequently, it appears that the position of the photoactivatable diazirine is not the prime determinant for labeling.

5) There is experimental evidence for specific cholesterol binding to a site containing E73 on VDAC1 (ref #21, Budelier et. al. 2017). While this paper is referenced to make a different point, cholesterol binding is not mentioned. The referenced paper maps three residues for a cholesterol binding pocket containing E73 on VDAC1 and shows that cholesterol binds to that pocket in both WT VDAC1 and E73Q VDAC1. The authors should reference the paper in context and consider the possibility that multiple lipids are binding to the same site. To ascertain whether cholesterol and ceramide bind to the

same site it would be useful to determine if excess cholesterol prevents pacCer labeling of VDAC1/2.

As outlined in our reply to comment #3 and described in the main text (p. 5, lines 19-24), we do consider that lipids other than ceramide can bind to the site containing the membrane-exposed Glu and now make a more explicit reference to the study by Budelier et al. (Ref. 46) regarding binding of cholesterol (p. 6, lines 20-25). However, our simulations and photolabeling experiments indicate that the Glu-containing site does not represent a major site of cholesterol binding. This can also be inferred from our new experiments, which demonstrate that C16-ceramide is not an effective competitor of pacChol labeling of VDAC1/2 under conditions where it causes a significant reduction in pacCer labeling (new Fig. 5b). It deserves mention that Budelier et al. performed all photolabeling experiments on detergent solubilized VDAC1. Moreover, the cholesterol analogue they used to label E73 was added in excess and carries the photoactive diazirine group in the aliphatic tail. In contrast, the cholesterol analogue used in our study has the diazirine located in the 6-position (new Supplementary Fig. 6). It is conceivable that these differences in experimental approach lead to divergent outcomes. Resolving this apparent discrepancy will require substantial additional work. In our opinion, this would go beyond the scope of the present study, which focuses on addressing the molecular principles by which ceramide can commit cells to death.

6) Ceramide inhibition of apoptosis is dependent on VDAC2: The data in Figure 4 show a clear role for VDAC2 in ceramide-induced apoptosis. The data in Figure 4 and supplementary Figure 8 also support that VDAC2 E84Q reduces ceramide-induced, VDAC2-dependent apoptosis. This represents the most important point of the paper (a specific ceramide binding site on VDAC2 is required for ceramide-induced apoptosis), but was only done in duplicate. The results show a small effect of E84Q on PARP1 cleavage and a more substantial effect on Casp3. These results (figure 8) would benefit from more replicates (with $n \geq 3$), error bars, quantitation and statistical analysis.

We now performed two additional rounds of experiments to verify the impact of the E84Q mutation in VDAC2 on ceramide-induced apoptosis. As shown in new Fig. 7b, this revealed that the reduction in PARP1 cleavage caused by the mutation is statistically significant, with a paired t-test P value of 0,0025 ($n=4$). Unfortunately, we ran out of the original stock of antibody against cleaved Casp3 and were unable to acquire a suitable replacement after screening multiple antibodies from different suppliers. Consequently, we were unable to expand the number of experiments aimed at detecting cleaved Casp3. However, as the two previous experiments in each case revealed a strong effect of the E84Q mutation on Casp3 cleavage (new Fig. 7c), we are confident that the impact of this mutation on ceramide-induced apoptosis is real.

7) In the molecular docking studies, why is the threshold for cholesterol set at 20% occupancy while the threshold for ceramide is set at 10% (figure 2d)?

Different thresholds were chosen for figure clarity, to compensate for the 6-fold larger amount of cholesterol compared to ceramide. A note has been added to the Methods section to clarify this (p. 18, lines 8-10). To illustrate the effect of the threshold, occupancy images of VDAC1 with and without ceramide are shown below at the same threshold for both lipids. At a 10% threshold for cholesterol occupancy even the membrane bulk, away from VDAC, is “occupied”. Note that even at the 10% threshold cholesterol cannot be seen to occupy the E73 binding site as much as ceramide does, despite being 6-fold more concentrated. Also note that at a 12.2% threshold there is still cholesterol presence away from VDAC, but no longer any in the vicinity of E73.

Threshold	VDAC1	VDAC1 in the absence of ceramide
10%		12.2%		
Reviewer #3

The authors combine coarse-grained computer simulations with functional measurements to show that the mitochondrial anion channel VDAC2 bind ceramide through a membrane-embedded glutamate residue. This finding suggests a role for VDAC2 in ceramide translocation through the membrane and subsequent apoptosis. I primarily reviewed the computational aspects of the paper. I like the synergistic use of simulations and experiments, which reveals interesting molecular details such as the likely ceramide binding sites. I have a number of mostly technical comments that should be clarified.

1) How was the VDAC2 model created? What is the sequence identity between the template mouse VDAC1 and VDAC2? If the sequence identity is lower than 90% or so then a tool such as MODELLER should have been used.

The mouse VDAC1 and 2 sequences have 74% identity in the barrel region (ignoring the alpha-helical, pre-barrel sequence). Structural homology is quite high, with barrel backbone residues at 1.7 Å RMSD, between the mouse VDAC1 structure (4C69) and the only available VDAC2 structure (from zebrafish; 4BUM). This supports the use of mouse VDAC1 as a structural template to build mouse VDAC2 on.

To build VDAC2, the coarse-grain structure of VDAC1 was mutated to the sequence of VDAC2 by a partially automated procedure. While this may seem a complex task to perform given the 60+ residue differences, it should be noted that in the Martini force field side-chain have only up to four particles, and that several residues have the same number of side-chain particles — meaning that changes from one to another need no structural modification, only a topological one. Another part of the mutations requires only side-chain particle deletions, which again is straightforward and automatable. In the few remaining cases that side-chain particles had to be constructed they could be simply appended to existing side-chain beads, with only particular care needed for mutations from Gly or Ala residues which, in the Martini forcefield, have no side-chain particles. This procedure has now been further clarified in the Methods section (p. 16, lines 9-16). This structural

mutation approach is possible because the Martini force field is robust to this magnitude of deviations from equilibrium structures. A simple steepest-descent energy-minimization is enough to bring all mutated side-chains close to equilibrium configurations.

2) How reliable are Coulombic interactions in CG FF such as Martini? This is important for assessing the interaction between E73 and ceramide and other lipids.

In general, the Martini force field only treats formal integer charges explicitly. This means that the side chain particle of a deprotonated Glu will have a -1 charge, as will the PO₄ particle of a phospholipid. Ceramide — as most lipid moieties apart from headgroups — has no such charges, and is Coulombic-wise invisible to E73. In the Martini model, however, particles can have a range of “polar” character, which defines preferential interactions with other particles of compatible polar character via Lennard-Jones-type (LJ) interactions. In this way the E73 charged side-chain particle, which is also of strong LJ polar nature, will readily bind polar moieties such as the ceramide head group, and all the more so in a membrane core environment of apolar particles that offer little competition.

Beyond this implicit encoding of charge/dipole interactions in LJ potentials, the Martini model has enough polarity types to resolve the differences we observed between ceramide binding to a deprotonated Glu (E73) or to a non-charged but polar Gln (E73Q) — or even to a protonated Glu; please see also our response to comment #4 below.

3) On line 81 "fine grain simulation analysis" is mentioned. "Fine grain" is jargon and it is not quite clear what the authors mean --- atomic resolution simulations?

The reviewer is correct to point out the use of jargon, and the term atomic resolution is now used instead.

4) What is the protonation state of E73 and how was it determined? What is the pKa of the residue? Do the results depend on it being protonated/deprotonated?

As the apparent pKa of E73 in VDAC1 is closely tuned to the physiological pH of the cytosol (pKa ~7.4; Bergdoll et al. 2017, Ref. 51), all main simulations were initially performed on VDACs with the membrane-buried Glu residue in the deprotonated state (see p. 17, lines 6-9). We now also repeated the simulations with this residue being protonated. In the Martini model, the protonated Glu is represented by a chargeless side-chain particle of polar nature. As shown in new Fig. 3, this markedly decreased the affinity of ceramide for its binding site. In agreement with the simulations, we also found that reducing the pH from 7 to 5 causes a marked reduction in the E73-dependent photolabeling of VDAC1 with pacCer (new Fig. 3c,d). These data are now described on p. 5 (lines 4-12) and p. 6 (lines 13-15) and imply that under stress-free conditions the membrane-buried E73 in VDAC1 is in its fully charged state at least a significant amount of time.

Interestingly, Bergdoll et al. (Ref. 51) found that acidification promotes association of two VDAC1 monomers into a dimer. Assembly of a high-affinity dimer relies on protonation of the membrane-facing Glu and likely involves hydrogen bonding between this residue and a serine (Ser43) at the dimer interface. VDAC oligomerization has been implicated as a component of the mitochondrial pathway of apoptosis (Refs. 49, 50) and may be part of the mechanism by which VDAC2 stabilizes the mitochondrial pool of Bax in response to apoptotic stimulation (Ref. 34). In view of our present findings, we speculate that binding of ceramide to the charged Glu residue may lower the threshold for VDAC oligomerization at neutral pH, thus potentially tipping the balance in VDAC2-mediated shuttling of Bax to trigger mitochondrial apoptosis. This concept is now outlined in the Discussion (p. 9, lines 20-27 and p. 10, lines 1-3) and provides a useful guide during our ongoing efforts to unravel how ceramides commit cells to death.

5) Does a E73D mutant retain ceramide binding?

We now also performed simulations of both VDACs with a deprotonated Asp at the Glu position and found that ceramide binding results were comparable to the wild-type cases. If anything, the shorter protrusion of the Asp side chain allows ceramide occupancy to reach closer to the VDAC backbone. These observations have now been included in new Supplementary Fig. 3 and are described on p. 5, lines 9-12.

6) Is $\epsilon=15$ standard for Martini? Given the importance of electrostatics in the current study, it should be clearly discussed what the known limitations of the CG FF are in this regard.

Indeed, 15 is the standard epsilon for Martini. It implicitly screens Coulombic interactions in the absence of explicit dipoles — namely those of water. It should be noted that for the charged side-chain of E73 the only significant charge interactions are those established towards the water-filled barrel core. As mentioned in our response to comment #2 above, lipid interactions of E73 occur via LJ interactions and are insensitive to the epsilon parameter.

7) MDAnalysis is not cited (only website given)

The reviewer is right to point out this oversight, which is now corrected by citing 10.1002/jcc.21787 (Ref. 65) and 10.25080/majora-629e541a-00e (Ref. 66).

Reviewers' Comments:

Reviewer #2:

Remarks to the Author:

I think this is a valuable paper that should be accepted for publication. I thank the authors for their thoughtful responses to our queries and criticisms and the additional experiments, which largely appear to strengthen the conclusions of the paper. I have a few remaining comments, that should be addressable with writing rather than experiments.

Comment #1:

I appreciate the point that pacCer photolabeling was used to identify VDAC 1 and VDAC 2 as potential ceramide binding proteins and that molecular simulation studies provide indirect evidence that E73 and E84 are part of a ceramide binding site. The author's assertion that identifying the photolabeled residues is beyond the scope of the paper is reasonable. I would note that the author's argument that site identification would not yield a meaningful result because it would require labeling in detergent is incorrect. Site identification does not require labeling in detergent. Labeling in a membrane environment using pacCer reconstituted in the membrane at <1% is entirely feasible. It is hard to equate how a solution concentration in detergent or bicelles translates to mole percent in a membrane. My guess is that 1 mole% is actually quite high. Membrane phospholipids are likely at molar concentration, and so <1 mole% would be mM. The lack of identification of the labeled residues does lead to an ambiguity of interpretation which should be addressed. pacCer may well be labeling the unprotonated side chain of E73/E84; the nucleophilic unprotonated side chain of an acidic amino acid is a likely site of insertion for an aliphatic diazirine, which may diffuse a short distance to find such a nucleophile. While the authors postulate, based on modeling data, that low pH and E73Q reduce pacCer labeling because they reduce ceramide binding, the effects of E73Q or low pH could be due to loss of the nucleophile, decreased binding, or both. I think they should indicate that they cannot be sure which interpretation is correct based on the experimental data, but that the modeling data suggests the glutamate side chain is important for binding.

#2

Thank you for your satisfactory response.

#3

Thank you for your satisfactory response.

#4

Thank you for your satisfactory response and for providing the additional figure.

#5

I am not entirely satisfied with the response, but I think the disagreement is not central to the major thrust of the paper and can be addressed with writing rather than experiments. First, the Budelier paper did not perform photolabeling in detergent-solubilized VDAC; labeling was performed in DMPC/CHAPSO bicelles. Second, two photolabeling reagents were used: KK174 with the diazirine in the aliphatic tail of cholesterol labeled E73. LKM38 with the diazirine at C7 (similar to the 6-azi-cholesterol used in the current paper) labeled T83 near E73. The possibility that pacChol may not label E73 (whereas other lipid photolabeling reagents do) may explain why removing the unprotonated glutamate (E73Q or low pH) does not reduce pacChol labeling.

The reviewer suggested assessing whether cholesterol prevented pacCer-labeling of VDAC, not whether ceramide prevented pacChol labeling. A plausible explanation for the observation that ceramide does not prevent pacChol labeling is that cholesterol or pacChol bind to the E73 site with much higher affinity than ceramide making it more difficult to compete.

I suggest that the authors correct their citation of the Budelier paper and simply acknowledge there is a discrepancy between the modeling data and photolabeling studies of cholesterol binding

to VDAC.

Respectfully submitted,
Alex S. Evers

Reviewer #3:

Remarks to the Author:

The author addressed my concerns. I am happy to see that the investigation of the charge state of E73 was fruitful.

I only have one minor correction:

1) Supplementary Fig 2: label "VDAC1" in lower left hand corner should probably be "VDACE73D"

RESPONSE TO REVIEWERS' COMMENTS

Reviewer #2

I think this is a valuable paper that should be accepted for publication. I thank the authors for their thoughtful responses to our queries and criticisms and the additional experiments, which largely appear to strengthen the conclusions of the paper. I have a few remaining comments, that should be addressable with writing rather than experiments.

Comment #1

I appreciate the point that pacCer photolabeling was used to identify VDAC1 and VDAC2 as potential ceramide binding proteins and that molecular simulation studies provide indirect evidence that E73 and E84 are part of a ceramide binding site. The author's assertion that identifying the photolabeled residues is beyond the scope of the paper is reasonable. I would note that the author's argument that site identification would not yield a meaningful result because it would require labeling in detergent is incorrect. Site identification does not require labeling in detergent. Labeling in a membrane environment using pacCer reconstituted in the membrane at <1% is entirely feasible. It is hard to equate how a solution concentration in detergent or bicelles translates to mole percent in a membrane. My guess is that 1 mole% is actually quite high. Membrane phospholipids are likely at molar concentration, and so <1 mole% would be mM. The lack of identification of the labeled residues does lead to an ambiguity of interpretation which should be addressed. pacCer may well be labeling the unprotonated side chain of E73/E84; the nucleophilic unprotonated side chain of an acidic amino acid is a likely site of insertion for an aliphatic diazirine, which may diffuse a short distance to find such a nucleophile. While the authors postulate, based on modeling data, that low pH and E73Q reduce pacCer labeling because they reduce ceramide binding, the effects of E73Q or low pH could be due to loss of the nucleophile, decreased binding, or both. I think they should indicate that they cannot be sure which interpretation is correct based on the experimental data, but that the modeling data suggests the glutamate side chain is important for binding.

We appreciate the reviewer's frankness and constructive criticisms. We now explicitly refer to the possibility that the unprotonated side chain of the membrane-buried Glu provides a site of insertion for the aliphatic diazirine in pacCer, which may diffuse a short distance to find such a nucleophile (p. 8, start of 2nd paragraph). However, two findings argue against the idea that pacCer labeling of VDAC1/2 is primarily driven by affinity of the aliphatic diazirine for the negatively charged Glu. First, pacCer labeling was progressively reduced by C₁₆-ceramide when added in 3- to 27-fold excess (Fig. 5b). Second, while pacCer and pacPC readily label VDAC1/2, pacPE and pacDAG do not, in spite of carrying the diazirine at the same position (Fig. 4, Suppl. Fig. 6).

Nevertheless, we acknowledge that our photolabeling experiments provide only indirect evidence for the binding of the ceramide head group to the Glu residue in VDAC1/2 as observed in the simulations. Regardless of the experimental conditions used, we still do not see how identification of the membrane-exposed Glu or a neighboring residue as the site of photo-labeling would enable us to validate or disprove the transient contacts between the ceramide head group and Glu residue predicted by the simulations. Introducing the photo-crosslinkable diazirine in the ceramide head would not be very meaningful, as this would likely perturb such contacts. Consequently, additional work and alternative approaches will be necessary to obtain definitive experimental proof for the ceramide head group – Glu interactions observed in the simulations.

Comment #2

Thank you for your satisfactory response.

Comment #3

Thank you for your satisfactory response.

Comment #4

Thank you for your satisfactory response and for providing the additional figure.

Comment #5

I am not entirely satisfied with the response, but I think the disagreement is not central to the major thrust of the paper and can be addressed with writing rather than experiments. First, the Budelier paper did not perform photolabeling in detergent-solubilized VDAC; labeling was performed in DMPC/CHAPSO bicelles. Second, two photolabeling reagents were used: KK174 with the diazirine in the aliphatic tail of cholesterol labeled E73. LKM38 with the diazirine at C7 (similar to the 6-azido-cholesterol used in the current paper) labeled T83 near E73. The possibility that pacChol may not label E73 (whereas other lipid photolabeling reagents do) may explain why removing the unprotonated glutamate (E73Q or low pH) does not reduce pacChol labeling.

The reviewer suggested assessing whether cholesterol prevented pacCer-labeling of VDAC, not whether ceramide prevented pacChol labeling. A plausible explanation for the observation that ceramide does not prevent pacChol labeling is that cholesterol or pacChol bind to the E73 site with much higher affinity than ceramide making it more difficult to compete.

I suggest that the authors correct their citation of the Budelier paper and simply acknowledge there is a discrepancy between the modeling data and photolabeling studies of cholesterol binding to VDAC.

We thank the reviewer for these remarks. We now corrected our reference to the Budelier paper, indicating that the latter study was performed on VDAC1-containing bicelles with cholesterol probes that carry the photoactive diazirine in the aliphatic tail or at C7. We also included a statement that additional work will be necessary to resolve the discrepancy between the simulations and photolabeling studies of cholesterol binding to VDACs (p. 8, first paragraph).

Reviewer #3

The author addressed my concerns. I am happy to see that the investigation of the charge state of E73 was fruitful. I only have one minor correction:

Supplementary Fig 2: label "VDAC1" in lower left hand corner should probably be "VDACE73D"

We thank the reviewer for spotting this error, which now has been corrected.